# Peptidyl nitroalkene inhibitors of main protease rationalized by computational and crystallographic investigations as antivirals against SARS-CoV-2

Francisco J. Medrano [1✉], Sergio de la Hoz-Rodríguez [2], Sergio Martí[3], Kemel Arafet[3], Tanja Schirmeister[4], Stefan J. Hammerschmidt [4], Christin Müller [5], Águeda González-Martínez[1], Elena Santillana[1], John Ziebuhr [5], Antonio Romero [1], Collin Zimmer[4], Annabelle Weldert [4], Robert Zimmermann[4], Alessio Lodola [6], Katarzyna Świderek [3], Vicent Moliner [3✉] & Florenci V. González [2✉]

The coronavirus disease 2019 (COVID-19) pandemic continues to represent a global public health issue. The viral main protease (M^pro) represents one of the most attractive targets for the development of antiviral drugs. Herein we report peptidyl nitroalkenes exhibiting enzyme inhibitory activity against M^pro ($K_i$: 1–10 μM) good anti-SARS-CoV-2 infection activity in the low micromolar range (EC$_{50}$: 1–12 μM) without significant toxicity. Additional kinetic studies of compounds **FGA145**, **FGA146** and **FGA147** show that all three compounds inhibit cathepsin L, denoting a possible multitarget effect of these compounds in the antiviral activity. Structural analysis shows the binding mode of **FGA146** and **FGA147** to the active site of the protein. Furthermore, our results illustrate that peptidyl nitroalkenes are effective covalent reversible inhibitors of the M^pro and cathepsin L, and that inhibitors **FGA145**, **FGA146** and **FGA147** prevent infection against SARS-CoV-2.

[1] Centro de Investigaciones Biológicas Margarita Salas (CSIC), Ramiro de Maeztu 9, 28040 Madrid, Spain. [2] Departament de Química Inorgànica i Orgànica, Universitat Jaume I, 12071 Castelló, Spain. [3] Departament de Química Física i Analítica, Universitat Jaume I, 12071 Castelló, Spain. [4] Institute of Pharmaceutical and BiomedicalSciences, Johannes Gutenberg-University Mainz, Staudinger Weg 5, 55128 Mainz, Germany. [5] Institute of Medical Virology, Justus Liebig University Giessen, Schubertstrasse 81, 35392 Giessen, Germany. [6] Dipartimento di Scienze degli Alimenti e del Farmaco, Università degli Studi di Parma, Parma, Italy. ✉email: fjmedrano@cib.csic.es; moliner@uji.es; fgonzale@uji.es

The impact of SARS-CoV-2 pandemic (COVID-19) has made the search for new therapies against coronaviruses urgent. The pandemic has resulted so far in over 600 million infections and over 6 million deaths worldwide, according to World Health Organization[1]. The evolution of SARS-CoV-2 virus has resulted in several highly contagious SARS-CoV2 strains that evade antibodies targeting the receptor binding domain and threaten the effectiveness of the current vaccines[2–4].

Among approved antivirals for the treatment of COVID-19 are nucleoside derivatives remdesivir[4,5] and molnupiravir[5,6], with uncertain efficacy for certain types of patients, and Paxlovid[6,7], a combination of the protease inhibitor nirmatrelvir and ritonavir, a metabolic booster that increase the effectiveness of the protease. Despite the remarkable efficacy of Paxlovid, it cannot be administered to patients with liver or kidney dysfunction. Furthermore, ritonavir, which blocks the rapid metabolism of nirmatrelvir by CYP3A, interacts in turn with other drugs limiting its use. These issues demonstrate that there is an urgent need to find new antivirals for SARS-CoV-2 and for other coronavirus outbreaks in the future.

During the replication cycle the coronavirus expresses two overlapping polyproteins, (pp1a and pp1b) and four structural proteins from the viral RNA[8]. In order to liberate the mature viral proteins required for replication, these polyproteins must be properly processed. There are two proteases coded in the viral genome, known as 3-chymotrypsin-like protease (3CLpro, 3CLP or nsp5, also termed main protease Mpro) and papain-like protease (PLpro or nsp3). Most of the cleavages are carried out by Mpro[9,10]. Both Mpro and PLpro are cysteine proteases with different site specificities.

Mpro is a three-domain cysteine protease essential for most maturation events within the precursor polyprotein[9–11]. The active protease is a homodimer. The active site is made up by a non-canonical Cys-His dyad located in the cleft between domains I and II[10–12]. Mpro hydrolyzes proteins predominantly between a P1 glutamine and a small P1′ amino acid, such as alanine, serine or glycine. For the P2 position, leucine is the most common amino acid in the sequence specificity for coronaviruses. There is no human homolog of Mpro which makes it an ideal antiviral drug target[13–16].

Peptidyl compounds have been shown to inhibit Mpro in in vitro assays and to prevent infection by SARS-CoV-2 in cell culture[17]. The warhead of these reported inhibitors are carbonyl groups [aldehydes[18], ketoamides[19], ketones[20]], nitriles (as marketed inhibitor nirmatrelvir)[21] and enoates[22]. Mpro inhibitors with new reactive warheads, chlorofluoroacetamide[23], dichloroacetamide[24], and nirmatrelvir analogs[25], have been designed with potent enzymatic inhibition and antiviral activity. In this context, the nitroalkene moiety has been previously reported by us as a valid warhead for inhibitors against cysteine proteases belonging to the papain family[26]. Irreversible inhibitors such as enoates can give rise to undesired side reactions. Alternatively, nitroalkenes represent Michael acceptor inhibitors following a reversible mode-of-action due to the low basicity of the nitronate intermediate[27]. Computational studies previously reported by us pointed to nitroalkenes as promising inhibitors of SARS-CoV-2 Mpro[28]. Interestingly, peptidyl nitroalkenes have been shown to act aspotent inhibitors of human cathepsin L (CatL) as we previously reported[27]. CatL has been also recognized as a potential target for the search of drugs against COVID-19 as it is found to enable viral cell entry by activating the SARS-CoV-2 spike protein by cleavage[29–32]. Dual inhibition against SARS-CoV-2 Mpro and human Cathepsin-L has been reported[33,34].

Analysis of interaction energies between the substrate (the peptide in the proteolysis reaction or the inhibitor in the case of the inhibition reaction) and the different binding pockets of SARS-CoV-2 Mpro based on multiscale quantum mechanics/molecular mechanics (QM/MM) studies, indicated that they are dominated by those in the P1:::S1 site[28,35,36]. However, the recognition portion dictates how the inhibitor is accommodated in the active site, which in turn affects the subsequent chemical reaction step. Consequently, the reactivity of the warhead and the favorable interactions between the recognition portion and the active site of the enzyme must be considered to design an efficient inhibitor[36]. In all, the experience accumulated based on the results derived from previous studies on this and other cysteine proteases can be used to guide the design of new compounds, and QM/MM simulations can be considered a useful tool to get a detailed description of the chemical steps of the inhibition of protein targets by covalent inhibitors.

Based on our previous studies on proteolysis reaction of the SARS-CoV-2 Mpro[35] and its inhibition mechanisms by peptidyl inhibitors with different warheads[28,35,36], we have designed and synthesized six peptidyl inhibitors with a nitroalkene warhead. These six inhibitors were able to inhibit the Mpro in vitro activity in the low micromolar range, and three of them were found to prevent SARS-CoV-2 infection in cell culture in the low micromolar range. In addition, these compounds were also tested against cathepsin L, a key enzyme for the viral entry into the cells. The crystal structures of Mpro in complex with these two most active inhibitors were solved to provide detailed information about the binding to SARS-CoV-2 Mpro. Molecular dynamics (MD) simulations with multiscale QM/MM potentials calculations were carried out to obtain the full free energy landscape of the inhibition reaction with the two most active inhibitors, confirming the interactions established with the active site residues of SARS-CoV-2 Mpro as well as their mechanism of action for the enzyme-inhibitor covalent complexes formation.

## Results

**Design and synthesis of the peptidyl nitroalkenes inhibitors**. Six inhibitors were designed and synthesized: three of them having the typical coronaviral protease glutamine surrogate (gamma-lactam) at the P1 site and a L-leucine at the P2 site (**FGA145**, **FGA146** and **FGA147**); the other three inhibitors display the typical cathepsin like inhibitors backbone: two having a L-homophenylalanine at the P1 site and a L-leucine at the P2 site (**FGA159** and **FGA177**), and one having a L-homophenylalanine at the P1 site and a L-phenylalanine at the P2 site (**FGA86**). The design of the inhibitors was based upon nitroalkene inhibitors reported by us[26,27] and previous Mpro inhibitors: compound **FGA146** is a direct analog of the Pfizer human SARS Mpro inhibitor PF-00835231[20], which is the progenitor for the clinically used inhibitor nirmatrelvir and **FGA147** is a direct nitroalkene analog of the well-established feline coronavirus protease inhibitor GC376[37].

For the synthesis of the ones having the glutamine surrogate at P1, the *N*-Boc protected amino alcohol **1** was prepared first as previously reported starting from L-glutamic acid[38]. The alcohol was then submitted to oxidation followed by a nitroaldol reaction with nitromethane in a one-pot procedure. The mixture of nitroaldols was then transformed into the corresponding inhibitors following a three-step sequence: Boc deprotection, peptide coupling and then dehydration through mesylate activation (Fig. 1). For the synthesis of the other three inhibitors having a homophenylalanine at the P1 site, *N*-Boc protected homophenyl alaninal was reacted with nitromethane and the nitroaldols were coupled with the corresponding peptide with free carboxylic terminus (Fig. 1, Table 1) (see Supplementary Experimental procedures for the preparation of the compounds).

**Fig. 1 Synthetic route for the preparation of the nitroalkene compounds used in this study.** Inhibitors FGA145, FGA146 and FGA147 were prepared through a common synthetic route starting from Boc-L-glutamic acid. FGA77, FGA86 and FGA159 were prepared through a synthetic route starting from Boc-L-homophenyl alaninal.

**Inhibition of the M$^{pro}$ activity by the peptidyl nitroalkenes**. The molecular structures and the $K_i$ values obtained for the inhibition of the M$^{pro}$ activity by the six peptidyl nitroalkene compounds are summarized in Table 1, respectively. For testing the inhibitory effect of the compounds, enzymes from two different expression systems were used (see Supplementary Enzymatic assays). An example of the enzymatic activity inhibition curves obtained from the enzyme using the pMal-M$^{pro}$ vector is shown in Supplementary Fig. 1. In Supplementary Fig. 2 the inhibition profiles for the six compounds obtained using the M$^{pro}$ from the expression using the pET21-M$^{pro}$ vector are shown. Compounds **FGA145**, **FGA146** and **FGA147** with a glutamate surrogate at the P1 site, leucine at P2 and an aromatic residue at P3 displayed inhibition in the low micromolar range, less than 10 μM. The N-terminal substitution of the aromatic residue is well tolerated which is shown through the substitution of the benzyloxycarbonyl (Cbz) group of **FGA147** by a 4-methoxy-1H-indole-2-carbonyl residue in **FGA146** leads to similar values of inhibition. Compounds **FGA86** with homophenylalanine at P1 and phenylalanine at the P2 sites, and an aromatic residue at P3; **FGA159** with a homophenylalanine at the P1 site, leucine at P2, and three more residues at P3, P4 and P5; and **FGA177** with a homophenylalanine at the P1 site, leucine at P2, and an aromatic residue at P3, also showed inhibition in the low micromolar

range. No irreversible character was observed over a 10-min period (Supplementary Fig. 1) denoting the compounds to be non-time dependent inhibitors as it was predicted by us[27,28]. The values of $K_i$ obtained using the plate reader assay and the continuous fluorometric assay were very similar, and the small non-significant differences might be due to differences in the enzymatic assay conditions, especially the different amount of the organic solvent DMSO (Table 1) (Supplementary Enzymatic assays).

**Cellular antiviral activity and cytotoxicity**. Three compounds (**FGA145**, **FGA146** and **FGA147**) were selected for the antiviral assay with infectious SARS-CoV-2. Huh-7-ACE2 cells were used for this antiviral assay. The antiviral activity and cytotoxicity assays are shown in Fig. 2. Compounds **FGA146** and **FGA147** showed potent antiviral activity against SARS-CoV-2 with EC$_{50}$ values in the low micromolar range (0.9 and 1.9 μM, respectively). **FGA145** showed less potent activity (EC$_{50}$ = 11.7 μM), in line with the enzymatic activity inhibition results (Supplementary Methods and Supplementary Data 4). The cytotoxicity of these compounds was very low with CC$_{50}$ values over 100 μM. To validate the therapeutic potential of the designed compounds, antiviral assays in suitable, primary cell systems (e.g., human bronchial epithelial cells) need to be done in the future.

**Table 1 Structure and inhibitory activity against SARS-CoV-2 M$^{pro}$ of the nitroalkene compounds.**

| Compound | 2D Structure | $K_i$ (µM)$^a$ | $K_i$ (µM)$^b$ |
|---|---|---|---|
| Nirmatrelvir | | 0.00315 ± 0.00042 | |
| FGA86 | | 17% inh. @ 20 µM | 2.67 ± 0.29 |
| FGA145 | | 3.71 ± 0.38 | 9.82 ± 1.50 |
| FGA146 | | 2.19 ± 0.18 | 0.96 ± 0.06 |
| FGA147 | | 2.18 ± 0.19 | 1.33 ± 0.05 |
| FGA159 | | 21.8 ± 1.5 | 1.19 ± 0.18 |
| FGA177 | | 25% inh. @ 20 µM | 8.59 ± 1.60 |

$^a$Data obtained using the assay with M$^{pro}$ obtained from the expression using the pMal-M$^{pro}$ vector.
$^b$Data obtained using the assays with M$^{pro}$ obtained from the expression using the pET21-M$^{pro}$ vector.

**Inhibition of other proteases.** While the cysteine protease M$^{pro}$ is inhibited with high potency by inhibitors **FGA145, FGA146** and **FGA147**, no inhibitory activity against the serine proteases: human matriptase (membrane-type serine protease 1, MT-SP1, prostamin) and bivalent expressed Zika Virus NS2B/NS3 (bZiPro) was observed for these compounds (Table 2) (Supplementary Methods). Besides inhibition of M$^{pro}$, compound **FGA145** was found to be a very potent inhibitor of cysteine proteases rhodesain

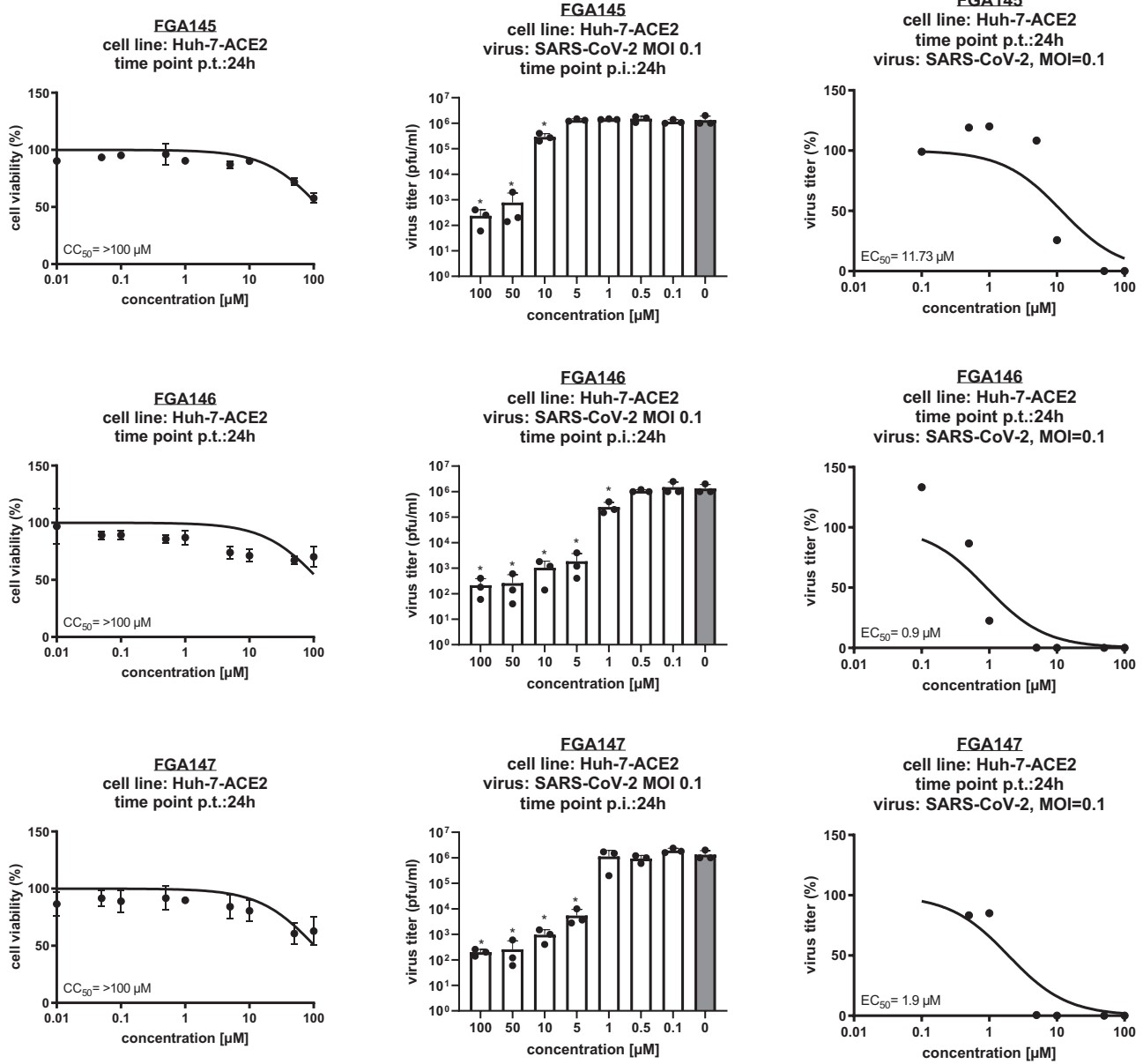

**Fig. 2 Inhibition of SARS-CoV-2 infection in Huh-7-ACE2 cell by FGA145, FGA146 and FGA147.** Cytotoxicity assays for the three compounds (left column), all of them presented a $CC_{50}$ greater than 100 μM. Effect of the three compounds on the virus titer (center and right columns); FGA146 was the most potent inhibitor with an $EC_{50}$ of 0.9 μM, followed by FGA147 and FGA145 with $EC_{50}$ of 1.9 and 11.7 μM, respectively. All graphs show means ± s.d.; asterisks indicate *p* values (*$p \leq 0.05$) obtained by two-tailed unpaired *t* tests.

(RhD), cruzain (CRZ), cathepsin L (CatL) and cathepsin B (CatB) with decreasing potencies from RhD to CatB (1.63 nM, 12.6 nM, 53.0 nM, 206 nM, respectively, Table 2). This finding is in line with previous reports of nitroalkenes as potent reversible inhibitors of these proteases[26]. Interestingly, all three compounds inhibit CatL, especially compound **FGA145**, denoting a possible multi-target effect of these compounds in the antiviral activity.

**Crystal structure of SARS-CoV-2 M^pro in the apo form and in complex with FGA146 and FGA147.** The SARS-CoV-2 M^pro in complex with **FGA146** crystallized in the P2₁ space group and diffracted up to 1.98 Å resolution (Supplementary Table 2) with one biological dimer in the asymmetric unit (Supplementary Data 1), and the complex with **FGA147** crystallized in the orthorhombic P2₁2₁2 space group and diffracted up to 1.62 Å resolution (Supplementary Table 2) (Supplementary Data 2) with

one monomer per asymmetric unit that forms the biological dimer with a crystallographic-symmetry related neighboring molecule (Supplementary Fig. 4). The protein structure can be subdivided into three domains (as shown in Supplementary Fig. 4), domain I and domain II containing the active site and domain III as the dimerization domain.

After the protein structures of the respective complexes were solved, significant electron density was found at the active site. This electron density could be unequivocally assigned to the corresponding inhibitors **FGA146** (Fig. 3a) and **FGA147** (Fig. 3b). Both inhibitors are covalently bound to the catalytic Cys145. A comparison of both inhibitors bound to the active site is shown in Fig. 3c.

The bound inhibitors, that mimic the natural peptide substrate, show a good geometric complementarity within the active site for subsites S1, S2 and S3 with the warhead located at the S1' subsite

| Table 2 $K_i$ values and selectivity towards some host proteases. | | | | | | |
|---|---|---|---|---|---|---|
| Compound | MT-SP1 | bZiPro | RhD (nM) | CRZ (nM) | CatL (nM) | CatB (nM) |
| FGA145 | n.i. | n.i. | 1.63 ± 0.22 | 12.6 ± 1.5 | 53.0 ± 4.1 | 206 ± 41 |
| FGA146 | n.i. | n.i. | n.d. | n.d. | 868 ± 60 | n.d. |
| FGA147 | n.i. | n.i. | n.d. | n.d. | 1993 ± 107 | n.d. |
| n.i.: no inhibition (<10%) was observed at a concentration of 20 µM. n.d.: not determined. All data are mean values ± standard deviation of three technical replicates. | | | | | | |

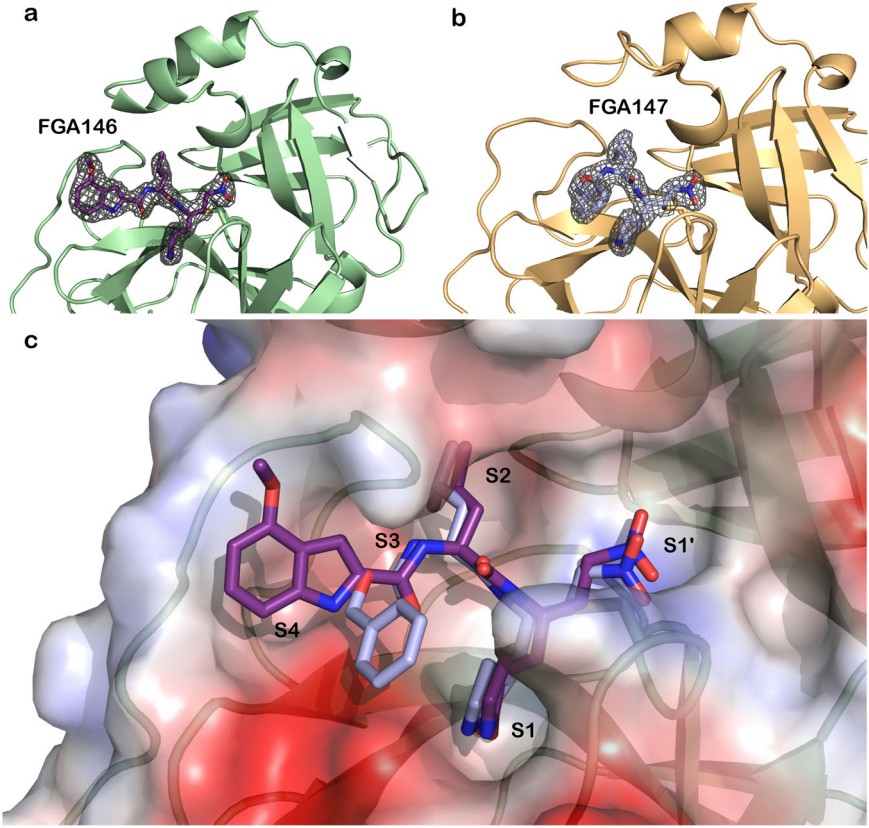

**Fig. 3 Crystal structures of SARS-CoV-2 M$^{pro}$ in complex with inhibitors.** M$^{pro}$ is shown in ribbon and the inhibitors in ball and stick representation. 2Fo-Fc electron density map contoured at 1σ (shown in gray mesh) of FGA146 (**a**) and FGA147 (**b**) bound covalently to the catalytic cysteine (Cys145). **c** Electrostatic surface representation of the active site of M$^{pro}$ with bound FGA146 (violet) and FGA147 (light blue). Red indicates negative charge and blue positive charge.

(Supplementary Fig. 5a–c). The nitro group of the warhead occupies, in both complexes, the "oxyanion hole" formed by the backbone amides of Gly143, Ser144, and Cys145 (Supplementary Fig. 5). The Sγ atom of the nucleophilic Cys145 forms a covalent bond to carbon C19 of the nitroalkene warhead of the inhibitor through a Michael addition (for numbering of the compounds see Supplementary Fig. 3). The S1 subsite is occupied by the glutamine surrogate γ-lactam ring that forms hydrogen bonds with the main chain of Phe140 and the side chain of Glu166 through the nitrogen atom (N16) of the ring, and to the side chain of His163 through the oxygen atom (O18) of the ring (Supplementary Fig. 5a–c and Fig. 6b, d). The carbon atoms of the side chain of these residues lie in a hydrophobic cavity defining the S1 subsite (Supplementary Fig. 5a–c and Fig. 6b, d). The second residue of both inhibitors is a Leu that is inserted into the hydrophobic S2 subsite made up by His41, Met49, and Met169 (Supplementary Fig. 5a–c). There are three hydrogen bonds between the main chain of the peptidyl inhibitor and the protein. They involve interactions between atoms N10 from the inhibitors and the backbone carbonyl of His164, N3 and OE1

from Gln189, and O1 and the backbone nitrogen from Glu166. The P3 residue side chain of the inhibitors, a methoxy indole carbonyl group in **FGA146** and a Cbz group in **FGA147**, shows different conformations when bound to the protein (Fig. 3c). In the **FGA147**, the Cbz group is oriented towards the solvent lacking interactions with the protein residues (Figs. 3c and S5a). However, the methoxy indole carbonyl group of **FGA146** is occupying the S4 subsite (Figs. 3c and S5a). This group forms a hydrogen bond between the N35 atom and the carbonyl atom from Glu166 (Supplementary Fig. 5b–e). This side chain is encased inside the S4 subsite formed by residues Glu166, Leu167, Pro168, Gln189, Thr190 and Ala191 (Supplementary Fig. 5d, e). The methoxy group is surrounded by the side and main chain atoms of residue Gln189, and by the main chain atoms of residues Thr190 and Ala191 (Supplementary Fig. 5d, e). The distance between the O atom of this methoxy group and the potential hydrogen bond partners is too far away and/or without a favorable geometry for this type of interaction (Supplementary Table 2), but it is enough to fix the position of this group and to orient the methyl group towards the solvent. Interactions between

the protein and inhibitor atoms with the distances between them are summarized in Supplementary Table 3. Besides the extensive hydrogen bond network, there are numerous non-polar interactions that contribute to the tight binding of the inhibitor.

The position of the nitro group of the warhead in the structure of M$^{pro}$ in complex with **FGA146** is not fixed by the interactions with the residues that form the "oxyanion hole" (Gly143, Ser144 and Cys145, Supplementary Fig. 7). In one of the monomers (mon. B), this nitro group forms hydrogen interactions with the N atoms from Gly143 and Cys145 (Supplementary Table 3, and Supplementary Fig. 7, protein in green and ligand in purple); while in the other monomer (mon. A), the nitro group moves away from the "oxyanion hole" and interacts with His41 (Supplementary Table 3, and Supplementary Fig. 7, protein in cyan and ligand in brown); showing a certain degree of flexibility upon binding to this S1 site. In the case of the inhibitor **FGA147**, the nitro group interacts with "oxyanion hole" (Supplementary Table 3) and, because there is only one monomer in the asymmetric unit, we observe only one conformation.

Concerning the protein, there are different conformations of some residues forming the active site. The most significant changes are located in the P2 helix, Ser46 to Asn51 (Fig. 4a) and in the P5 loop, Asp187 to Ala193 (Fig. 4a). The first segment is the α-helix that takes part in the formation of the S2 binding subsite. In the structure of the **FGA146** complex, this segment is displaced towards the inside of the active site with respect to the structure of the **FGA147** complex, consequently widening the S2 subsite (Fig. 4a). The second segment is the P5 loop, at this subsite only the inhibitor **FGA146** was observed to be bound, while the P3 side chain of **FGA147** is pointing towards the solvent. This segment is tightly packed around **FGA146** fixing its conformation. **FGA147** does not bind to this subsite in our structure, and the residues are positioned further away (Fig. 4a). Also, there is a small difference at the P4 β-hairpin flap. There is one loop close to the active site with a different conformation (Fig. 4a, loop I), and another loop further away from the active site also with a different conformation (Fig. 4a, loop II).

Besides the differences observed for the active site located in domains I and II, there are additional differences in domain III between the structures of the M$^{pro}$ in complex with **FGA146** and **FGA147** (Fig. 4b). The superposition of these structures shows clear displacements in the positions of four of the helices (α-6, α-7, α-8 and α-9) in monomer A, as a consequence the loops linking those helices are also displaced.

These changes could not be observed in monomer B where all the helices from this domain are very well aligned with only small non-significant differences (Fig. 4b). Upon observing these differences in only one monomer we superposed monomer A and monomer B from the M$^{pro}$ in complex with **FGA146**, shown in Fig. 4c. Here, we were able to observe the same changes that occur in monomer A between the **FGA146** and **FGA147** structures (Fig. 4b). This indicates a great flexibility of the M$^{pro}$ with no coordinated changes in both monomers. These changes could be due to the different space groups in which the protein crystallized, as stated previously[39] or just might indicate the flexibility of this protein which has to accommodate itself to be able to catalyze the proteolysis of the polyprotein to liberate the mature proteins essential for the virus replication.

**Computational study of the SARS-CoV-2 M$^{pro}$ inhibition by FGA146 and FGA147.** Based on the results derived from the kinetic studies presented in previous section, the inhibition reaction was studied according to the general mechanism proposed in Fig. 5 with the two most promising inhibitors: **FGA146**

and **FGA147**. After the binding between the inhibitor and the enzyme to form the non-covalent E:I complex, the first chemical step would involve the activation of Cys145 by a proton transfer to His41 which can take place concomitantly with the nucleophilic attack of the sulfur atom of Cys145 to the C19 atom of the inhibitor to form an intermediate, **E-I$^{(-)}$**. Then, the reaction is completed by the transfer of the proton from the protonated His41 to the C20 atom of the inhibitor to render the final **E-I** covalent adduct.

The free energy profiles derived from free energy surfaces (FESs) of the SARS-CoV-2 M$^{pro}$ inhibition with both **FGA146** and **FGA147** obtained by M06-2X/6-31 + G(d,p)/MM MD simulations (see Supplementary Figs. 12, 13; Tables 4, 5, 6 and Computational Methods) confirm that the activation of Cys145 takes place concertedly with the inhibitor-enzyme covalent bond formation, **E:I** to **E-I$^{(-)}$** step. Interestingly, we already observed this concerted activation and nucleophilic attack of Cys145 when exploring the acylation step of the proteolysis reaction[35], but previous inhibition processes explored in our laboratory have rendered stepwise processes where the activation of the Cys145 precedes the covalent formation between the sulfur atom of Cys145 and the different tested warheads of the inhibitors[28,36]. The last step of the inhibition reaction corresponds to the proton transfer from His41 to the C20 atom of the inhibitor, **E-I$^{(-)}$** to **E-I** step.

The chemical steps of the inhibition process are exergonic in both cases (−15.6 and −9.8 kcal·mol$^{-1}$ with **FGA146** and **FGA147**, respectively), and the activation of free energies, determined by the formation of the intermediate covalent intermediate, **E-I$^{(-)}$**, are almost equivalent (15.3 and 15.6 kcal·mol$^{-1}$ with **FGA146** and **FGA147**, respectively).

Structures of the different states appearing along the reaction optimized at M06-2X/MM level are shown in Fig. 6, while a list of key inter-atomic distances obtained on the representative stable states is listed in Supplementary Tables 7 and 8. The estimation of the main interaction energies between residues of SARS-CoV-2 M$^{pro}$ and the inhibitors **FGA146** and **FGA147** computed in the **E:I** and in the **E-I** states are shown in Supplementary Fig. 15.

**Thermal stability of M$^{pro}$ in the absence and presence of the inhibitors.** We have examined the thermal stability of the protease in the absence and in the presence of the inhibitors using circular dichroism (CD). The $T_m$ value for M$^{pro}$ did not change, significantly, in the presence of **FGA177** (0.1 ± 0.2 °C in the range of 25 to 100 μM, Figs. 7c and S8d). In the presence of **FGA86** (Figs. 5c and S8a) at the lower concentration (25 μM) there was a slight increase in stability (0.9 ± 0.4 °C) and at the higher concentration (100 μM) there was a slight decrease in stability (−1.7 ± 0.4 °C). The presence of the inhibitors **FGA145** (−1.1 ± 0.2 °C at 25 μM and −3.7 ± 0.1 °C at 100 μM, Figs. 5c and S8b) and, while **FGA146** (−4.2 ± 0.2 °C at 25 μM and −6.4 ± 0.1 °C at 100 μM, Fig. 7a, c) and **FGA147** (−3.0 ± 0.1 °C at 25 μM and −4.2 ± 0.2 °C at 100 μM, Fig. 7b, c) lead to a significant decrease of the $T_m$ value. It has been reported that the association of covalently-bound compounds induce shifts of $T_m$ values to lower temperatures with an apparent destabilization of the protein[40–42]. Thus, these data agree with the formation of a covalent bond between the protein and compounds **FGA86, FGA145, FGA146** and **FGA147**. Based on the X-ray crystallographic analyses and thermal stability data, we can conclude that these last four compounds bind covalently to the protein. The inhibitor **FGA159** showed an increase in stability of 2.5 ± 0.2 °C at a concentration range between 5 and 100 μM (Figs. 5c and S8c).

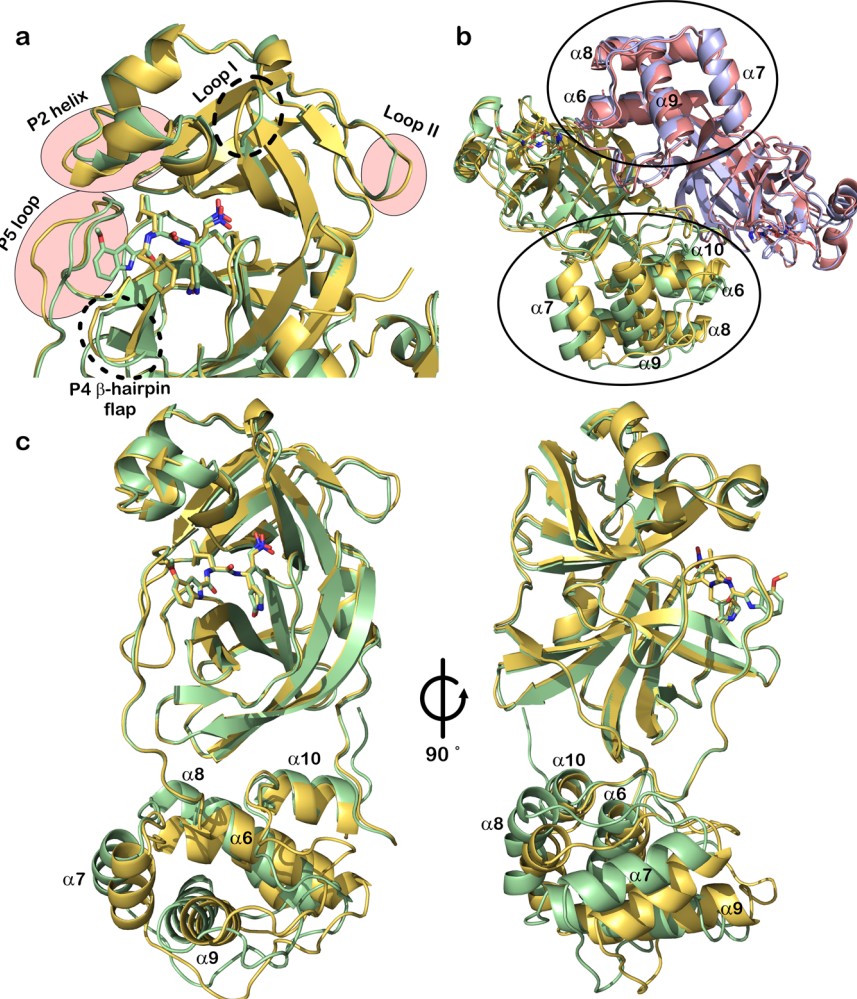

**Fig. 4 Conformational changes in M^pro upon binding of the inhibitors. a** Most significant changes in the active site and domain I are located at P2 helix and the P5 loop. Smaller changes can be observed at the P4 β-hairpin flap, loop I and loop II. **b** Superposition of M^pro in complex with FGA146 and FGA147. Monomer A and B from the complex with FGA146 are shown in light green and yellow orange, respectively, and from the complex with FGA147 are shown in salmon and light blue, respectively. Significant displacements of some of the helices from the dimerization domain (Domain III, circles) of monomer A can be observed, while the same domain from monomer B does not show these displacements. **c** Superposition of monomer A (light green) and B (yellow orange) of the M^pro in complex with FGA146. The helical dimerization domain (Domain III) shows differences in the relative positions of some of the helices. Four of them show significant displacements (α-6, α-7, α-8 and α-9) while the las helix (α-10) does not show any significant displacement.

**Fig. 5 Proposed Mechanism of SARS-CoV-2 M^pro Cysteine Protease Inhibition by nitroalkene compounds.** The mechanism has three steps: entry of the inhibitor into the active site, then addition of the cysteine thiolate, and finally protonation of the nitronate.

**Binding of inhibitors to M^pro.** Figure 8a shows the binding isotherms of **FGA145** and **FGA146** to M^pro by ultra-centrifugation, and Fig. 8b shows the binding of **FGA147** to M^pro measured by ITC (Supplementary Data 5). The measured binding dissociation constants for these compounds were in the low micromolar range, the $K_d$ for **FGA145** was $11.8 \pm 1.05$ μM,

the $K_d$ for **FGA146** was $7.28 \pm 0.58$ μM (the average obtained from using both, absorption and fluorescence, data), and the $K_d$ for **FGA147** was $2.86 \pm 0.25$ μM. For all three inhibitors we obtained a stoichiometry of one, which is compatible with binding of one molecule of the inhibitor to the active site.

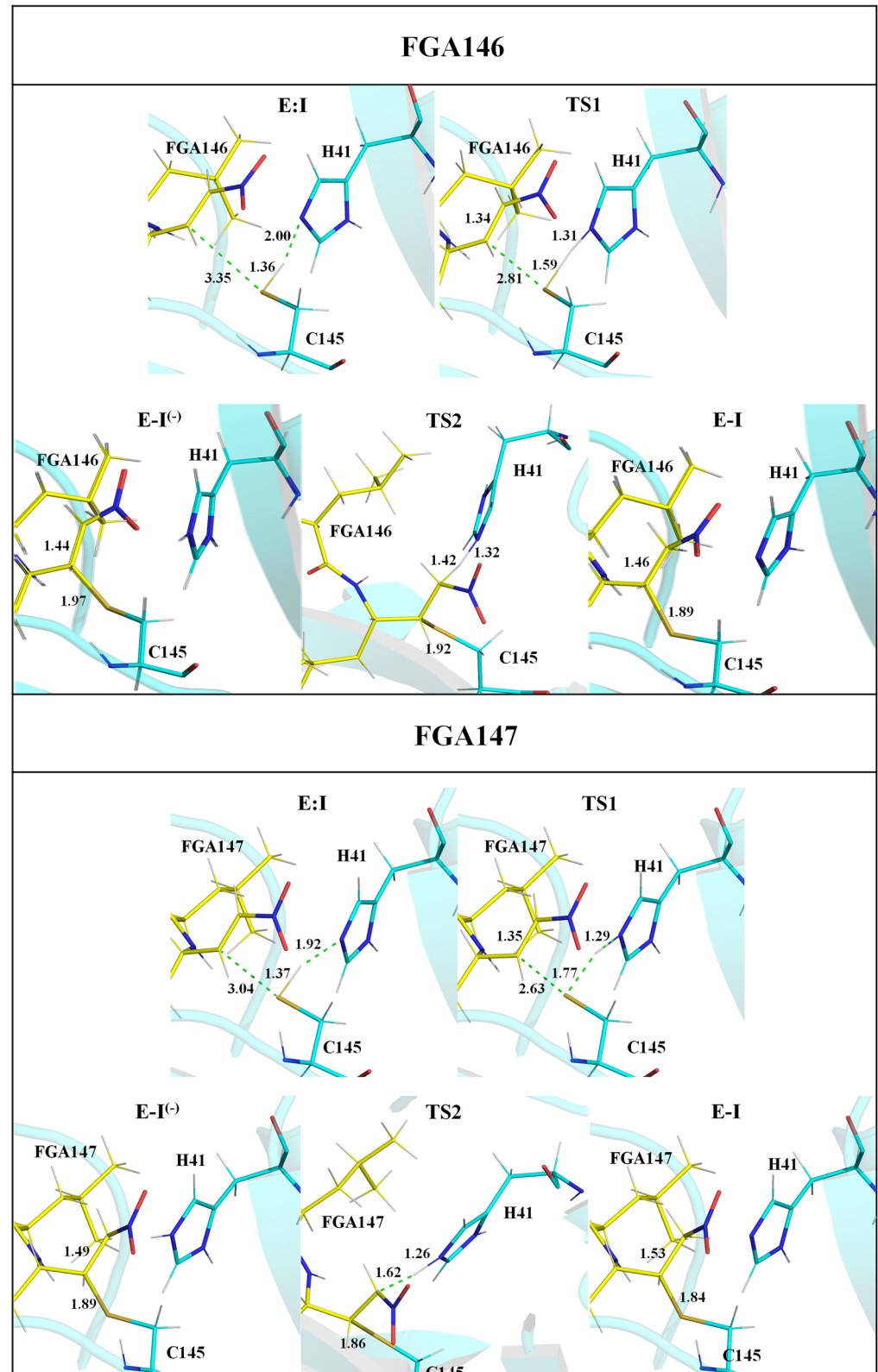

**Fig. 6 Structures of the states. Detail of the M06-2X/6-31 + G(d,p)/MM optimized structures of the states located along the inhibition reaction of M$^{pro}$ by FGA146 (top panels) and FGA147 (bottom panels).** Carbon atoms of the inhibitor are shown in yellow, and those of the catalytic residues C145 and H41 are shown in cyan. Key distances are in Å.

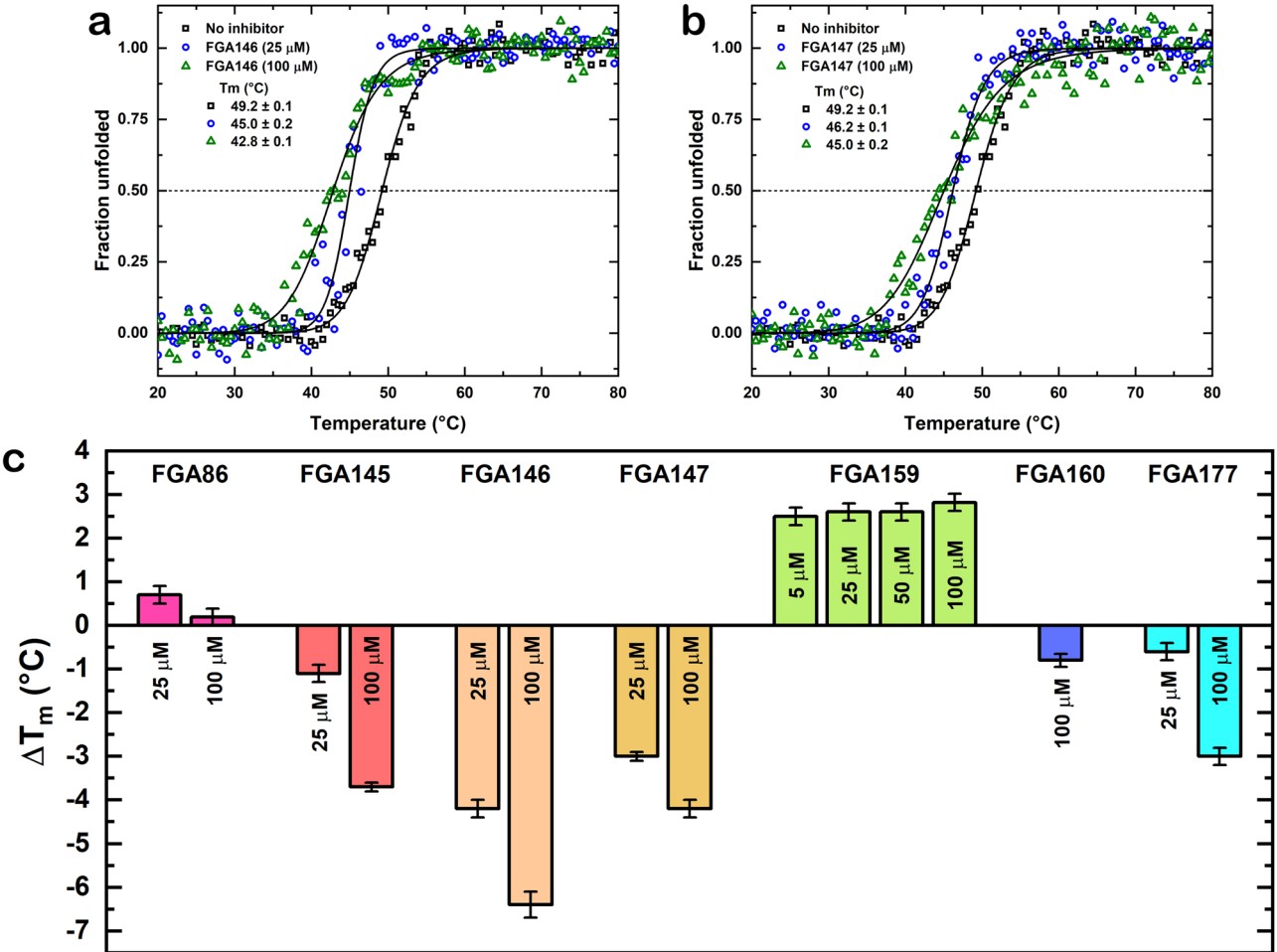

**Fig. 7 Effect of the inhibitors on the thermal stability of M$^{pro}$. a** Thermal stability of M$^{pro}$ in the presence of FGA146 using circular dichroism. **b** Thermal stability of M$^{pro}$ in the presence of FGA147 using circular dichroism. The $T_m$ value of M$^{pro}$ in the absence of inhibitors (black squares) was 49.2 °C, while in the presence of 25 (blue circles) and 100 μM (green triangles) of FGA146, the values decreased to 45.0 and 42.8 °C, respectively. In the case of FGA147, these values decreased to 46.2 and 45.0 °C, respectively. **c** Change of the $T_m$ in the presence of the inhibitors. FGA86 shows a slight increase in stability at the lower concentration (25 μM) and a decrease in stability at the higher concentration (100 μM). There is no significant change in stability in the presence of FGA177; while in the presence of FGA145, FGA146 and FGA147 shows a significant decrease in stability, suggesting that they could bind covalently to the protein. The presence of FGA159 increased the stability of the protein. The error bars represent the fitting error of the denaturation data.

**Reversibility**. The results of the experiments carried out by dilution to check for the reversibility of the binding of the inhibitor **FGA146** to M$^{pro}$ are shown in Fig. 9 (Supplementary Data 6). The activity of the M$^{pro}$ preincubated with **FGA146** seems to recover over time (shown in blue in Fig. 9), which makes it likely that the compound inhibits its target reversibly. For the reversible control Nirmatrelvir (nitrile warhead) there is only partial recovery of activity and for the irreversible control (vinyl sulfone warhead) there is no visible recovery of activity.

**Pharmacokinetic assays**. A pharmacokinetic profile has been performed for most active inhibitor **FGA146** including physico-chemical parameters (Supplementary Table 9), cysteine reactivity (Supplementary Figs. 20 and 21), stability (Supplementary Table 10 and Figs. 22, 23 and 24), dilution assay, inhibition assay (Supplementary Fig. 25), DNMT2 MST-displacement assay (Supplementary Fig. 26), logP determination by HPLC (Supplementary Table 11, Figs. 27, 28) and permeability (Supplementary Figs. 29 and 30).

**Discussion**

We have designed, synthesized and measured the inhibitory effect of a series of peptidomimetic compounds with a nitroalkene

warhead on the enzymatic activity of M$^{pro}$ and cell infection. We have also examined the possibility of using a nitroalkene warhead that due to its reversible binding[26] should decrease the possibility of side effects due to unwanted reactions with other cellular components.

Six compounds (**FGA86, FGA145, FGA146, FGA147, FGA159** and **FGA177**) were prepared in good yields via a short and straightforward synthetic route. Among the inhibitors we have synthesized there are three similar to inhibitors already reported, they have the same peptidomimetic moiety but they differ in the warhead. **FGA145** is similar to two previously reported inhibitors[43]. **FGA146** is similar to PF-00835231[20] and **FGA147** is similar to GC376[44], the rest of the inhibitors present a new molecular structure. All of them exhibited enzyme inhibitory activity ($K_i$: 110 μM) and three of them (**FGA145, FGA146** and **FGA147**), having the typical coronaviral protease glutamine surrogate (gamma-lactam) at P1 site and a L-leucine at P2 site, gave good anti-SARS-CoV-2 infection activity in the low micromolar range (EC$_{50}$: 112 μM) without significant toxicity. Additional kinetic studies of the selectivity of **FGA145, FGA146** and **FGA147** show that they are also potent inhibitors of cathepsin L (CatL), revealing a multitarget effect.

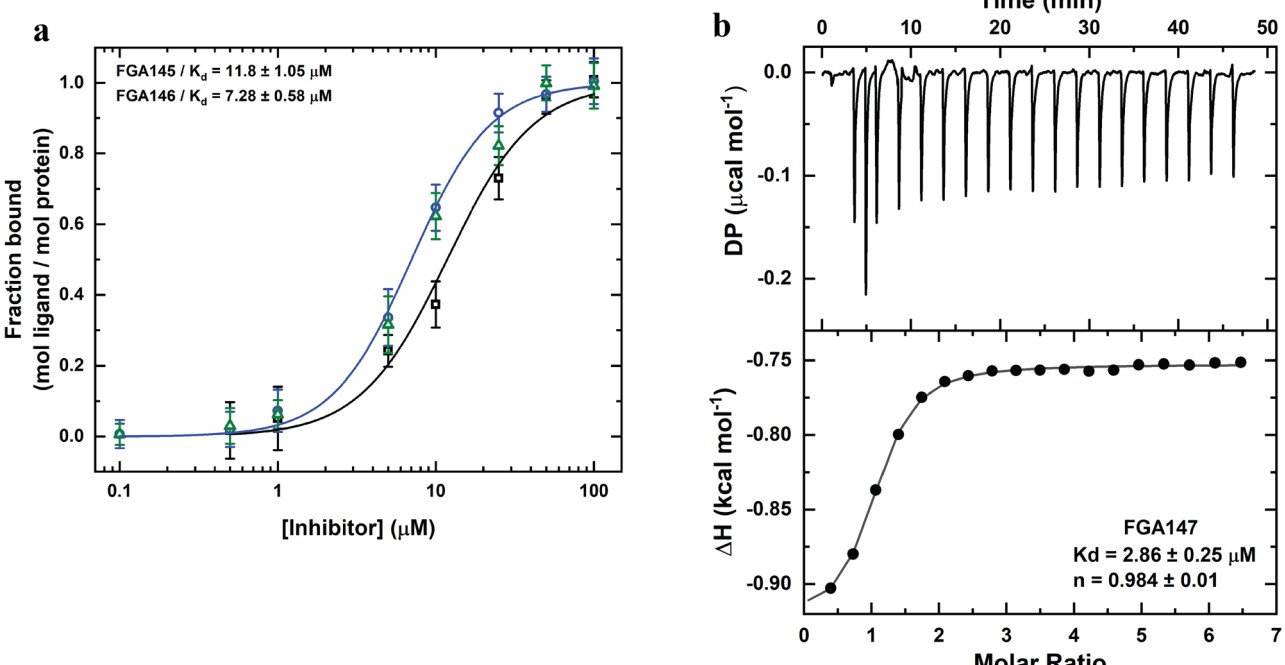

**Fig. 8 Binding of inhibitors to M^pro. a** Binding isotherm of FGA145 (black squares) and FGA146 (blue circles and green triangles) to M^pro. The concentration of FGA145 was measured by absorption, and that of FGA146 was measured by absorption (blue circles) and fluorescence (green triangles). All data are mean values ± standard deviation of two technical replicates. **b** ITC binding profile of FGA147 to M^pro.

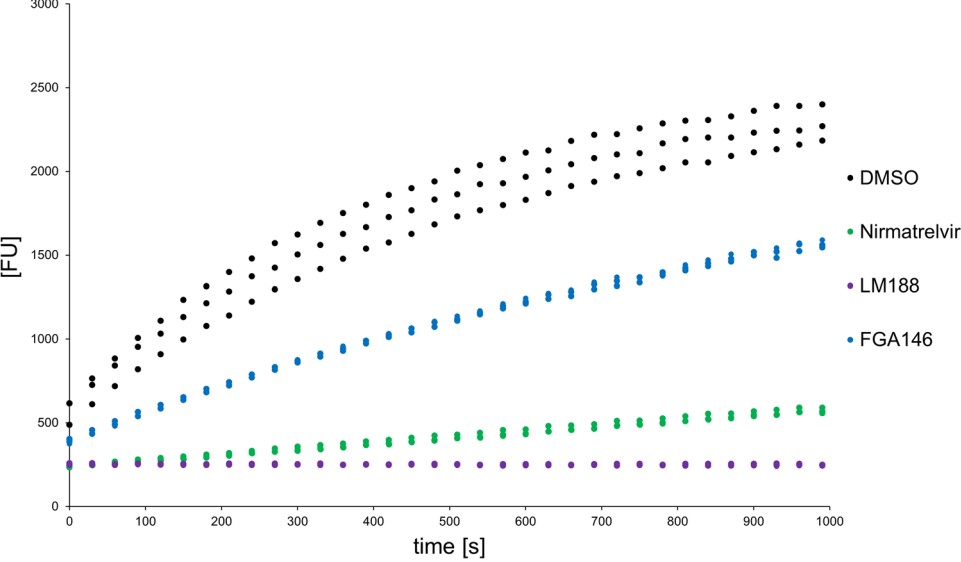

**Fig. 9 Reversibility of the binding of the inhibitor FGA146 to M^pro.** In black a DMSO control with no inhibitor added is shown, in green Nirmatrelvir as a reversible control and in blue the compound FGA146. All samples were measured as triplicates. Due to substrate depletion over time only the first 1000 s are shown.

The crystal structures of the M^pro in complex with **FGA146** and **FGA147** were solved and confirmed the binding modes. The covalent inhibitory character of these inhibitors is similar to other peptidomimetic inhibitors[18,19,22]. Our crystal structures, that virtually overlap with the structures derived from the computer simulations that were initiated from a previously crystallized M^pro in complex with a different inhibitor (N3), the structures corroborate the great conformational flexibility of the dimer of M^pro. Flexibility of the active site has been reported to be needed to accommodate the different natural cleavage sites present in the polyprotein of SARS-CoV-2, as well as some

conformational flexibility of other regions of the protein (Domain II and III)[39]. The crystal structures of M^pro in complex with **FGA146** and **FGA147** show a very similar binding mode to that of the similar inhibitors PF-00835231 and GC376. The only significant difference is the warhead and its binding to the "oxyanion hole" (Supplementary Fig. 9).

Through thermal denaturation we have been able to observe that the inhibitors might stabilize or destabilize the protein, in some cases a destabilization of more than 6 °C was found. The analysis of the crystal structures together with these thermal denaturation data shows the influence of the inhibitors on the

whole structure of M$^{pro}$. This instability induced by active site inhibitors might be exploited to increase their potency against the virus replication if they could be combined with inhibitors that bind to sites other than the active site to further disrupt the activity of this essential protease for the virus.

Finally, QM/MM computer simulations assisted in elucidating the way of action of the most promising compounds, **FGA146** and **FGA147**, by generating the complete free energy landscape of the inhibitor-enzyme covalent complex formation. The results of the inhibitory mechanism, that appear to be equivalent in both cases, suggest that, after the non-covalent enzyme:inhibitor binding step, the **E:I** formation, the activation of Cys145 takes place concertedly with the inhibitor-enzyme covalent bond formation. In the second chemical step, the final proton transfer takes place from His41 to the Cα atom of the inhibitors, E-I$^{(-)}$ to E-I step. The resulting free energy profiles for the covalent inhibition of SARS-CoV-2 M$^{pro}$ with **FGA146** and **FGA147** show how the processes are exergonic in both tested inhibitors, which determined not only by the favorable binding step but by the stability of the final adduct, **E-I**, with respect to the initial E:I complex (see Supplementary Computational Methods).

In summary, we have designed and synthesized six compounds as covalent reversible inhibitors of the SARS-Cov-2 M$^{pro}$ with the nitroalkene warhead. The three most active inhibitors were active at low micromolar concentrations against SARS-CoV-2 and did not show significant toxicity. These compounds were also active against human Cathepsin L in the nanomolar range, denoting a dual activity. The crystal structures, aided by computer simulations, of the two most promising inhibitors in complex with M$^{pro}$ show the mechanism of action of these inhibitors and the interactions established between the inhibitor and the protein.

## Methods

**General procedure for the preparation of nitroalkenes.** To a stirred solution of alcohol **1** (1.05 g, 4 mmol) in dichloromethane (32 mL) was added Dess-Martin periodinane (1.82 g, 4.3 mmol) and sodium bicarbonate (361 mg, 4.3 mmol). The resulting mixture was stirred at room temperature for 1 h. Then the reaction mixture was cooled with an ice-bath and triethylamine (0.17 mL, 1.21 mmol) and nitromethane (1.33 mL, 24.4 mmol) were. Then the mixture was stirred for 15 h at room temperature and then was quenched with a saturated aqueous solution of NH$_4$Cl (10 mL), the mixture was extracted with CH$_2$Cl$_2$ (3 × 15 mL) and the combined organic layers were washed with HCl 1 M, then with a saturated aqueous solution of sodium bicarbonate and then dried over Na$_2$SO$_4$. Then the solvent was evaporated and the residue was purified by column chromatography silica gel, CH$_2$Cl$_2$/MeOH (99:1 to 85:15) to afford the desired product as a yellow oil (71%).

The corresponding nitroaldol (0.73 mmol) was dissolved in dichloromethane (2.1 mL) and placed in an ice-bath. Then trifluoroacetic acid (1.1 mL) in dichloromethane (1.1 mL) was added dropwise and the mixture was stirred at room temperature for 3 h. The reaction mixture was evaporated in vacuo to give the product as a colorless solid. The resulting mixture was submitted to the next step without any further purification.

To a solution of the ammonium trifluoroacetate (0.80 mmol) and the carboxylic acid (0.89 mmol) in dichloromethane (8 mL) cold with an ice-bath, HOBt·H$_2$0 (121 mg, 0.89 mmol) was added. After 15 min at the same temperature, DIPEA (0.56 mL, 3.23 mmol) was added dropwise. After another 15 min, EDC (186.2 mg, 0.97 mmol) was added and the mixture was stirred for 16 h at room temperature. Then the mixture was quenched with saturated ammonium chloride solution (10 mL) and extracted with dichloromethane (3 × 20 mL). The combined organic layers were washed with HCl 1 M, with a saturated aqueous sodium

bicarbonate solution and then dried over Na$_2$SO$_4$. Then the solvent was evaporated and the residue was purified by column chromatography silica gel, CH$_2$Cl$_2$/MeOH (100:0 to 85:15) to afford the desired product (64%, two steps).

To an ice bath cold solution of peptidyl nitroaldol (0.66 mmol) in dichloromethane (6.6 mL) was added DIPEA (0.24 mL, 1.39 mmol), then methanesulfonyl chloride (0.056 mL, 0.73 mmol). The resulting mixture was stirred overnight, then it was quenched with a saturated aqueous solution of NH$_4$Cl (10 mL) and extracted with dichloromethane (3 × 20 mL). The combined organic layers were washed with HCl 1 M then with a saturated aqueous sodium bicarbonate solution and then dried over Na$_2$SO$_4$. Then the solvent was evaporated and the residue was purified by column chromatography silica gel, CH$_2$Cl$_2$/MeOH (99:1 to 9:1) to afford the desired product (68–81%) (NMR Spectra of all compounds in Supplementary Data 3).

For the preparation of all the compounds, the coupling steps and nitroalkene formation were done following the experimental procedure detailed above.

For the preparation of compound **FGA159**, the hydrolysis and hydrogenation steps were done following standard experimental procedures.

*Cloning of M$^{pro}$ gene.* The M$^{pro}$ gene was cloned in two different vectors with a similar strategy. First, the sequence of the gene coding for M$^{pro}$ (nsp5) SARS-CoV-2 was optimized for *Escherichia coli* expression, synthesized and cloned directly into a pUCIDTKan. The M$^{pro}$ gene was amplified from the vector pUCIDTKan-M$^{pro}$ and cloned into the vector pET21a (Novagen) named pET21-M$^{pro}$. Second, the M$^{pro}$ gene was inserted into the pMal plasmid harboring the C-terminal hexahistidine-tagged sequence of SARS-CoV-2 Mpro named pMal-M$^{pro}$ (Prof. John Ziebuhr, Justus Liebig University Gießen, Germany). The sequence contained the native nsp4/nsp5 M$^{pro}$ cleavage site between MBP and Mpro as well as the native nsp5/nsp6 cleavage site between M$^{pro}$ and the hexahistidine tag, thus enabling the purification of native Mpro.

*Protein expression and purification.* **SARS-CoV-2 M$^{pro}$** The vector pET21-M$^{pro}$ was transformed into *E. coli* Tuner (DE3) cells (Novagen, Merck, Madrid, Spain). These cells were grown in 2xYT medium supplemented with ampicillin (100 mg/L) at 37 °C. The expression was induced by the addition of 0.1 mM isopropyl-β-D-thiogalactopyranoside (IPTG), and let to grow for an additional 16 h. The cells were harvested by centrifugation and resuspended in the lysis buffer. The cells were lysed by sonication and the supernatant was then loaded onto a HisTrap FF column (GE Healthcare). The fractions containing the protease were then pooled, and PreScission protease containing a hexahistidine tag was added. The mixture was then dialyzed and the PreScission-treated M$^{pro}$ solution was applied to a HisTrap FF column to remove the PreScission protease, the C-terminal tag, and M$^{pro}$ with uncleaved hexahistidine tag. The processed M$^{pro}$ was collected in the flow-through and concentrated to 10 mg/mL. The expression of SARS-CoV-2 M$^{pro}$ using the vector pMal-M$^{pro}$ was performed exactly as described previously[45]. ***Human matriptase.*** Recombinant expression and purification were mainly performed as described previously[46]. The pQE30 plasmid, containing the human matriptase (membrane-type serine protease 1, MT-SP1, prostamin) was kindly provided by Prof. Torsten Steinmetzer (Philipps University Marburg, Germany). Since MT-SP1 is expressed as inclusion bodies, no leakage suppression was needed and, hence, the plasmid was transformed in *E. coli* BL21-Gold (DE3) (Agilent Technologies, Santa Clara, CA, USA) cells. After growing them in LB medium supplemented with ampicillin (100 mg/mL) to an OD$_{600}$ of 0.6–0.8, overexpression was induced

by addition of 1 mM IPTG over night (o.n.) at 20 °C. Cells were harvested by centrifugation, flash frozen in liquid $N_2$ and stored at −80 °C until further usage. For protein refolding and purification from inclusion bodies, cell pellets were resuspended in lysis buffer (50 mM TRIS−HCl pH 8.0, 300 mM NaCl, 10% (v/v) glycerol and 1 mM β-ME), supplemented with lysozyme and DNase and stirred for 1 h at room temperature (rt). After that, cells were further lysed by sonication (Sonoplus HD 2200; Bandelin, Berlin, Germany) and again centrifuged. The supernatant was discarded and the pellet was washed with lysis buffer. Proteins were solubilized in a denaturing solubilization buffer (50 mM TRIS−HCl at pH 8.0, 6 M urea, 10% (v/v) glycerol and 1 mM β-ME) by stirring o.n. at rt. The suspension was again centrifuged to remove cell debris. The supernatant was subjected to IMAC on a HisTrap HP 5 ml column (Cytiva Europe GmbH, Freiburg im Breisgau. Germany), using IMAC buffer A (50 mM TRIS−HCl pH 8.0, 6 M urea, 20 mM imidazole and 1 mM β-ME) in a linear gradient with IMAC buffer B (50 mM TRIS−HCl at pH 8.0, 6 M urea, 250 mM imidazole and 1 mM β-ME). The fractions, containing eluted MT-SP1 were refolded by a 2-step dialysis over 12 h each at 4 °C in dialysis buffer A (50 mM TRIS−HCl at pH 9.0, 3 M urea and 1 mM β-ME) and anion exchange (IEX) buffer A (50 mM TRIS−HCl at pH 9.0, 1 mM β-ME) prior to IEX chromatography on a Resource Q 1 ml column (Cytiva Europe GmbH, Freiburg im Breisgau. Germany), using IEX buffer A in a linear gradient with IEX buffer B (50 mM TRIS−HCl at pH 9.0, 1 M NaCl and 1 mM β-ME). Eluted MT-SP1 was flash frozen in liquid $N_2$ and stored at −80 °C. *Zika Virus 2 NS2B_{CF}/NS3_{pro}*. The bivalently expressed ZIKV protease was expressed and purified as described previously[47]. Briefly, the pETDUET vector containing bZiPro (purchased from Addgene) was transformed into competent *E. coli* BL21 Gold (DE3) cells (Agilent Technologies, Santa Clara, CA, USA) and grown in LB medium containing ampicillin at 37 °C until they attained an optical density ($OD_{600}$) of 0.8. Overexpression was induced o.n. by addition of 1 mM IPTG at 20 °C. After harvesting, cells were flash frozen in liquid $N_2$ and stored at −80 °C until protein purification. Herein, cell pellets were resuspended in lysis buffer (20 mM TRIS−HCl at pH 8.0, 300 mM NaCl, 20 mM imidazole, 0.1% (v/v) Triton X-100, RNase, DNase, lysozyme and 1 mM DTT) and lysed by sonication. After centrifugation, bZiPro from the cleared supernatant was purified by IMAC on a HisTrap HP 5 ml column with a step-gradient of washing buffer (20 mM TRIS−HCl at pH 8.0, 300 mM NaCl and 20 mM imidazole) and elution buffer (20 mM TRIS−HCl at pH 8.0, 300 mM NaCl and 250 mM imidazole). The eluted fractions, containing bZiPro were subjected to a gel filtration step (HiLoad 16/600 Superdex 75; GE Healthcare, Chicago, IL, USA) in SEC buffer (50 mM TRIS−HCl at pH 8.0 and 150 mM NaCl). Eluted bZiPro was flash frozen in liquid $N_2$ and stored at −80 °C. *Cruzain*. Cruzain (CRZ) was kindly provided by Dr. Avninder S. Bhambra (De Montfort University, Leicester, UK). *Cathepsin L, Cathepsin B*. Both cathepsin L (CatL) and cathepsin B (CatB) were purchased from Calbiochem (Merck Millipore, Burlington, Massachusetts). *Rhodesain*. Rhodesain (RhD) was recombinantly expressed and purified as reported previously[48].

*Enzymatic assays.* Proteolytic activity was determined by cleavage of fluorescence resonance energy transfer peptide substrates. The inhibitor was added into the M^pro solution in the reaction buffer, mixing and allowing the mixture to equilibrate for 10 s, and then initiated by adding the Dabcyl-KTSAVLQ ↓ SGFRKME-(Edans)-Amid substrate solution. Compounds in Table 1 were diluted in DMSO (**FGA145, FGA146** and **FGA147**), N,N-dimethylformamide (**FGA86** and **FGA177**) and ethanol (**FGA159**). Due to the deleterious effect of the solvents on the activity of M^pro and for

consistency of the data, the concentration of solvent was kept constant at 1% (v/v) in all experiments. All measurements were made in triplicates. For measurements using the TECAN Infinite F200 PRO plate reader each well was composed of 180 μL buffer, 5 μL enzyme in buffer, 10 μL inhibitor in DMSO or ethanol, and 5 μL substrate in DMSO (measuring conditions for all the proteases are summarized in Supplementary Table 1). The amount of solvent in these experiments was 7.5%.

**FGA146**, the indole harboring compound, was the only one revealing a strong fluorescence in this assay at higher concentrations. To rule out that bleaching of this fluorescence interferes with our readout by overlaying the fluorescence increase caused by the enzymatic substrate cleavage, control measurements were performed. Therefore, the assay was repeated without addition of substrate, hence, the negative slope due to bleaching of **FGA146** was determined. The relative activity values were then corrected for the negative slope of each inhibitor concentration (Fig. S1).

*Cell-based antiviral activity and cytotoxicity assays.* Huh-7 cells that overexpress human angiotensin-converting enzyme 2 (ACE2) (Huh-7-ACE2; kindly provided by Friedemann Weber (Institute of Virology, Justus Liebig University Giessen)) were grown in Dulbecco's modified Eagle's medium supplemented with 10% fetal bovine serum and antibiotics (100 μ/mL penicillin and 100 μg/mL streptomycin) at 37 °C in an atmosphere containing 5% $CO_2$. The SARS-CoV-2 isolate Munich 929[31] was kindly provided by Christian Drosten (Institute of Virology, Charité-Universitätsmedizin, Berlin). Cytotoxic concentrations 50% ($CC_{50}$) of the compounds used in antiviral activity assays were determined using MTT assays as described previously[49]. To determine effective concentrations 50% ($EC_{50}$) of the respective compounds, Huh-7-ACE2 cells were inoculated with SARS-CoV-2 at a multiplicity of infection (MOI) of 0.1 plaque-forming units (pfu) per cell. After incubation for 1 h at 33 °C, the virus inoculum was replaced with fresh cell culture medium containing the test compounds at the indicated concentration. After 23 h at 33 °C, the cell culture supernatants were collected and virus titers were determined by virus plaque assay as described previously[50].

*Crystallization data collection and structure determination.* Crystallization trials were performed at 295 K using the sitting-drop vapor-diffusion method with commercial screening solutions using a Cartesian Honeybee System (Genomic Solutions, Irvine, USA) nano-dispenser robot.

For data collection, crystals were cryo-protected with 30% (v/v). X-Ray data collection experiments were performed at the ALBA Synchrotron (Cerdanyola del Vallès, Spain) BL13 XALOC beamline, and at the ESRF Synchrotron (Grenoble, France) ID30B beamline. Data were indexed and integrated, scaled and merged using XDS[51]. The structures were solved by molecular replacement using the previously reported SARS-CoV2 M^pro structure (PDB: 7K3T) with Molrep[52]. The inhibitor molecule was added manually. Data processing and refinement statistics are listed in Supplementary Table 2.

*Circular dichroism.* Circular dichroism measurements were carried out on a JASCO J-720 (Jasco, Tokyo, Japan) spectropolarimeter equipped with a Peltier type temperature controller and a thermostatized cuvette cell linked to a thermostatic bath. Spectra were recorded in 0.1 cm path length quartz cells with a response time of 4 s and a band width of 2 nm. The protein concentration used was 0.15 mg/mL in 20 mM Tris-HCl buffer at pH 7.5 and 100 mM NaCl. The observed ellipticities were converted into the molar ellipticities [θ] based on a mean molecular mass per residue of 110.45 Da. Thermal denaturation experiments were performed by increasing the temperature from 20 to

80 °C at 30 °C/h. Tm represents the temperature at the midpoint of the unfolding transition. The CD signal was followed at 230 nm and the concentration of organic solvent was kept constant at 2.5%. Two concentrations of each compound were used (25 and 100 μM).

*Binding measurements.* For direct binding measurements we used the sedimentation method previously reported[53]. A solution of known concentrations of protein and ligand was centrifuged in 1 mL polycarbonate tubes in an MLA-15 rotor in the Optima MAX-XP ultracentrifuge (Beckman) at 150,000 rpm (700,200 x *g*) for 4 h at 25 °C. At the end of the centrifugation, the lower half of the tube contained the protein in equilibrium with free ligand; the upper half contained only free ligand and essentially no protein, as checked by control measurements. The upper 0.5 mL was withdrawn carefully, and the free ligand was measured. The bound ligand concentration was taken as the difference between total concentration of ligand added to the tube and the measured free concentration. The values of the binding equilibrium constant and the number of sites were obtained by fitting the binding equation to the data. **FGA145** and **FGA146** were the only two ligands with an absorption spectrum that allowed us to measure its concentration spectrophotometrically. The higher concentrations were measured directly and for the lower concentrations the solution has to be concentrated to obtain absorption spectra with sufficient intensity. Also, **FGA146** was measured fluorometrically due to the fact that this compound present intrinsic fluorescence. The protein concentration used in the experiments was 14.8 μM, and the concentration range used for **FGA145** was 0.5 to 100 μM, and for FGA146 was 0.1 to 100 μM. The experiments were performed in 20 mM TRIS-HCl buffer at pH7.5 containing 0.1 M NaCl and 1% DMSO. Isothermal Titration Calorimetry (ITC) titrations were performed in a PEAQ-ITC calorimeter (Malvern, Westborough, MA, USA), using a M$^{pro}$-containing solution of 250 μL at 29.5 μM in 20 mM TRIS-HCl buffer at pH 7.5 containing 0.1 M NaCl and injections at 150 s intervals of 2 μL ligand-containing solution (up to adding 36.4 μL) at 25 °C and 750 rpm. The ligand solution consisted of a concentration of 1 mM of the compound **FGA147** containing 5% of DMSO. A fitted offset parameter was applied to account for potential background. Data processing was performed using the MiroCal PEAQ-ITC Analysis software. None of the other compounds were completely soluble with 5% DMSO at the concentration required for the ITC experiments.

*Reversibility experiments. Dilution assays.* Buffer: 20 mM Tris, 0.1 mM EDTA, 1 mM DTT, 200 mM NaCl pH = 7.5. For the dilution assay Nirmatrelvir[21] was used as a reversible control and LM188[54] as an irreversible control (Fig. 10).

All compounds were preincubated with the enzyme (10 μM) at a concentration of at least 20 times the IC$_{50}$ (70 μM for LM188 and 76.4 μM for FGA146). For Nirmatrelvir a (relatively) higher concentration (10 μM) was chosen to ensure the full inhibition of the enzyme. The incubation was done at RT for 30 min in assay buffer. DMSO was used as a control. After incubation the Enzyme/Inhibitor (or DMSO) mixtures were diluted 100-fold in assay buffer. The substrate (Dabcyl-KTSAVLQ ↓ SGFRKME-Edans) was added with a final concentration of 25 μM, and the fluorescence was measured over 1 h at 25 °C as triplicates.

**QM/MM simulations.** After setting up the molecular models, the reaction was studied using a QM/MM approach from the equilibrated structures. The QM subset of atoms includes the P1' and P1 positions of the inhibitor, together with C145 and H41 residues of

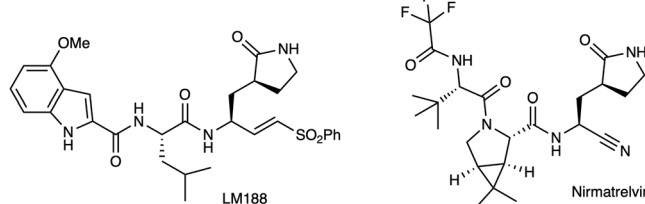

**Fig. 10 Chemical structure of LM188 and Nirmatrelvir.** Both were used as controls in the reversibility experiments: inhibitor LM188 was used as an irreversible inhibitor and Nirmatrelvir as a reversible inhibitor.

the protein. Four link atoms were inserted where the QM/MM boundary intersected covalent bonds in the positions indicated on Supplementary Fig. 11. Thus, QM part consisted of 57 atoms for both inhibitors. All the calculations were performed with the QMCube suite[55], for which the combination of the OpenMM and Gaussian09[56] programs was used for constructing the potential energy function. The AMBER ff03[57] and the TIP3P[58] force fields were selected to describe the MM atoms, and the Minnesota functional M06-2X[59] with the split-valence 6–31 + G(d,p) basis set[60] were used to treat the QM subset of atoms. This functional has been tested and shown to be suitable for modeling this type of reactivity[27,28,35,36,61–64]. The position of any atom over 20 Å from the substrate was fixed to speed up the calculations.

The reaction mechanisms for each inhibitor were initially explored using the nudged elastic band[65] approach to set up plausible starting geometries for the transition structures. The information obtained in this stage was used in the fine-tuning of the calculation of the FES, in terms of potential of mean force (PMF). The PMF for each chemical step was obtained using the combination of the umbrella sampling (US) approach[66] with the weighted histogram analysis method[67].

Finally, the protein:inhibitor interaction energy was estimated as a contribution of each residue of the protein, computed as an average along QM/MM MD simulations. The interaction energy is decomposed in a sum over residues provided that the polarized wave function ($\Psi$) is employed to evaluate this energy contribution, and the QM sub-set of atoms were described by the semiempirical Hamiltonian AM1[68] in these QM/MM MD calculations. A detailed description of the computational methods can be found in the Supplementary Material.

**Reporting summary.** Further information on research design is available in the Nature Portfolio Reporting Summary linked to this article.

## Data availability

Data are available in the main text, the Supplementary Information, the Supplementary Data files 1–6, and the Zenodo repository at: https://zenodo.org/uploads/10066288. The atomic coordinates and structure factors have been deposited into the Protein Data Bank with accession codes 8BGA and 8BGD. Materials and all other data are available from the corresponding authors on reasonable request.

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

## Acknowledgements

This research was funded by the Consejo Superior de Investigaciones Científicas, grant number PIE-202020E224, the Spanish Ministerio de Ciencia e Innovación (ref. PID2021-123332OB-C21 and PID2019-107098RJ-I00), the Generalitat Valenciana (PROMETEO with ref. CIPROM/2021/079, and SEJI/2020/007), Universitat Jaume I (UJI-B2020-03, UJI-B2021-71 and SomUJIcontracovid crowdfunding campaign). K.Ś. thanks to Ministerio de Ciencia e Innovación and Fondo Social Europeo for a Ramon y Cajal contract (Ref. RYC2020-030596-I). The authors wish to thank the staff of beamlines ID30B (ESRF Synchrotron) and BL13-XALOC (ALBA Synchrotron) for their generous and much appreciated support, and the Serveis Centrals d'Instrumentació Científica of Universitat Jaume I for technical support. The work was also supported by a Research Grant of the University Medical Center Giessen and Marburg (UKGM, to C.M.), the von Behring-Röntgen-Stiftung (project 71_0016, to C.M.) the Deutsche Forschungsgemeinschaft (DFG, project 530813989, to C.M.). Finally, the authors acknowledge the computer resources at Mare Nostrum of the Barcelona Supercomputing Center (QH-2022-2-0004 and QH-2022-3-0008), as well as the local computational resources founded by Generalitat Valenciana - European Regional Development Fund (REF: IDIFEDER/2021/02).

## Author contributions

All authors contributed to this work. A.R. and F.V.G. conceived and designed the study; F.J.M. obtained the crystal structures and performed the CD experiments; A.G.-M. performed the kinetic experiments; E.S. performed the cloning; S.d.l.H.-R. performed the synthesis; S.J.H. performed protein expression, purification, and enzymatic assays of M^pro and off-targets; C.M. and J.Z. performed the virus and cell assays; C.Z., A.W. and R.Z. performed the pharmacokinetic assays; S.M., K.A. and A.L. carried out the computer simulations; All authors participated in the discussion of the results; F.J.M., F.V.G., V.M., A.L., K.Ś., A.R. and T.S. participated in writing the original draft. All authors have read and approved the published version of the manuscript.

## Competing interests

The authors declare no competing interests.
