## [Peer Review File · Communications Chemistry]

Reviewers' comments:

Reviewer #1 (Remarks to the Author):

This article by Medrano et al. reports both experimental and computational results concerning the development of covalent inhibitors of the main protease (MPro) of SARS-CoV-2, which is a validated target for the treatment of the COVID-19 disease. Since protease inhibitors that have been approved as antiviral agents (e.g., nirmatrelvir) present various limitations, there is an ongoing interest in the development of new compounds as noticed by the authors. Basing on previous findings of dipeptidyl nitroalkenes as potent reversible inhibitors of cysteine proteases (ref. 23), this work demonstrates now that tailored peptidomimetic molecules equipped with the nitroalkene group form covalent adducts with the Cys145 thiol group at the active site of MPro and act as competitive in-vitro inhibitors of this enzyme. The best compounds (FGA145, FGA146 and FGA147) are characterized by inhibition constants KI and dissociation constants KD in the low micromolar range. Two crystallographic structures of Mpro in complex with FGA146 and FGA147 are reported that provide valuable structural information. These results together with other data (cytotoxicity assays, binding isotherms and ITC, off-target inhibition studies) should support peptidyl nitroalkenes as promising candidates in future studies aimed to find potent (nM) Mpro inhibitors. Nevertheless, it turns out also that further work is clearly necessary to elucidate the reversible (or irreversible) mode of action of the Mpro inhibitors as well as to determine the actual significance of the computational results in comparison with the experimental data. More particularly, additional data and an improved description of the computational protocol are necessary to give sufficient methodological detail for the reproducibility of the simulations. Overall, I could recommend the publication of a revised version of the manuscript provided that the authors take into account and/or give reasonable answers to the following specific questions and comments:

- In the “Design and synthesis ...” subsection, the authors merely indicate that they designed six inhibitors. In the revised manuscript, they should describe the guidelines followed (or tools employed) in the molecular design of the inhibitors.
- The authors claim in the abstract and in the Conclusions that their peptidyl nitroalkenes are reversible inhibitors of the Mpro enzyme. This reversibility is highlighted as important to reduce possible side effects associated to the inhibitors. However, in contrast with the approach followed in ref. 23, in the present work, there is no experimental verification of the reversibility of the inhibition via dialysis assays or dilution assays showing how fast the MPro activity is recovered. Such experimental proof should be incorporated to the revised manuscript.
- The Methods section of the paper reproduces the Supporting Information (perhaps this is a formatting or uploading issue). A much shorter Methods section would be adequate for the main text. Note that there are two different Tables in the SI numbered as S2.
- Figure 10 displays two binding isotherms for compounds FGA145 and FGA146, and one ITC profile for FGA147. However, the details of the ITC experiments or ultracentrifugation are not included in the Methods section so that they must be added to the revised paper. It is unclear why the authors selected ultracentrifugation to analyze the binding of FGA145 and FGA146 to Mpro, while ITC is used to

characterize the binding of FGA147. Hence, it would be highly convenient to compare the binding affinity of the three inhibitors using the same methodology, preferably ITC. Perhaps technical (solubility?) issues may complicate the ITC experiments, but the availability of the ITC-based free energies for FGA146-7 would be a valuable complement to the crystallographic structures.

- Considering that the reported compounds inhibit the Mpro activity in the low micromolar range, the authors should most likely point out in the Discussion that further molecular design and optimization will be required to reach the nanomolar inhibition displayed by other inhibitors like nirmatrelvir. The usage of the adjective “potent” in the Abstract does not seem adequate.

- The discussion of the computational results is quite confusing in general. For example, it is mentioned (p 19, l 408) that the activation of Cys145 by His41 takes place concertedly with the bond formation between the inhibitor and the enzyme, but the structural evidence supporting this claim is missing. Similarly, it is emphasized (p 19, l 406) that the “complete” free energy landscape of the inhibitor-enzyme covalent complex formation is generated, but the step of non-covalent binding ($E+I \rightarrow E:I$) is not investigated. It is stated that FGA146 and FGA147 are kinetic and thermodynamically “indistinguishable” in their binding to Mpro (p 19, l 412) and, in the same paragraph, that the different P3 fragment in the two compounds has a significant effect in their reactivity (p 19, l 419). The authors also consider that the predicted “thermodynamic behavior of FGA146 and FGA147 agrees with the very close experimentally determined K_i values”. However, the schematic free energy profile in Figure 2 shows that the Mpro-FGA146 adduct is 5.8 kcal/mol more stable than Mpro-FGA147, which corresponds to a 10000-fold difference in binding affinity! It is not clear either how the supposed reversibility of the inhibitors can be compatible with the relatively-large free energy barriers for the reverse process (25-31 kcal/mol; Figure 2). Overall, these and other observations (see below) suggest that the actual significance of the computational results is scarce.

- The presentation of the energetic and structural data produced by the simulations should be improved to provide a clear description of the reaction mechanism. For example, the schematic free energy profiles in Figure 2 should be replaced by (or at least augmented with) the corresponding PMF plots including the data points associated to each window (i.e., not just the interpolation curve). This energetic information should be complemented with molecular models showing the most relevant features of the intermediate and TS-like configurations as well as with the corresponding plots showing the evolution of the reactive distances along the reaction coordinate s (I note in passing that a proper definition of s within the space expanded by the reactive distances is missing). In the supplementary material, Table S6 collects key inter-atomic distances for the initial E:I and the final E-I states, but other structures along the reaction coordinate like the E-I(-) intermediate and the TS-configurations should be included, identifying also the origin of such distances (average values from M062X/MM PMF windows? M062X/MM optimized structures?). Table S2 and S6 (including interatomic distances between heavy atoms) may be merged to facilitate the comparison between X-ray and computational models.

- The crystallographic 8BGA and 8BGD structures display the flexibility of several structural motifs, which delineate the active site region (P2-helix in the S2 subsite, P5 loop, P4 flap,...) This is a particularly relevant observation that is in line with former assessments of other X-ray structures of MPro (e.g., ref 33). Consequently, the ability of the MD simulations to shed light onto this flexibility and/or to compare

with the X-ray models will require the sampling of the slow backbone motions over hundreds of nanoseconds. In the present work, the X-ray models of the E-I complexes are just compared with a “single” QM/MM MD structure (Figure 7). Although the settings of the QM/MM MD calculations (presumably, AM1/MM MD) remain essentially unknown (see below), the length of such simulations seems likely to be below 2-5 nanoseconds given that only 1000 MD frames are used for averaging the interaction energies (Figure 3). Hence, two important problems with the simulations arise at this point: (i) the enzyme loop motions are undersampled; (ii) there is no proper analysis of the MD data in order to select and characterize statistically-sound representative structures to be compared with the X-models. Sufficiently-extensive and unconstrained simulations in explicit solvent using classical force fields followed by well-designed clustering analysis, RMSD calculations, ... may be helpful.

- Figures S13 and S14 can be removed because they do not add any relevant information with respect to Figure 3. In addition, the legend of Figure 3 needs to be improved by including the description of items a), b), c), and d). In the legend of Figure S12, include the name of the other inhibitors, the corresponding PDB codes and the criteria used for the superposition of the different structures. The interaction between the phenyl group in P3 of FGA147 with Gln189 and Glu166 (p 16, I 336) that explains its orientation is not shown in Figure 3. Again, clustering calculations are required here to characterize quantitatively the conformations adopted by the P3 residue in FGA146 and FGA147. A representation of the most populated clusters should be shown in Fig. 8 while the statistical abundance of the inhibitor...oxyanion hole contacts should be reported.

- The Comm Chem guidelines for referees remark the importance that authors provide enough data or methodological information to help others replicate their work. Although the reproducibility of complex simulations is not trivial, the lack of technical details/settings and relevant structures in the supporting material make things worse. For example:

What QM level of theory is used to obtain the RESP charges (AM1-bcc, HF/631G*,...)? Which version of GAFF is used?

Which is the minimum distance between solute atoms and the solvent box walls?

How do the RMSD plots look like?

How are the end structures for the NEB calculations selected or built? Which are the NEB settings (number of images, spring constants, ...)?

How many critical points are located on the static reaction paths at the M062X/MM level? Which are their properties and geometries? How many points are obtained in the MEP calculations?

How many windows are considered in the PMF calculations? Is there an equilibration period within each window? Which criteria are followed to assess the convergence of the PMF profiles with the total simulation time?

What is the exact definition of the collective variable adopted for the PMFs?

The computational bottleneck of QM/MM simulations is usually due to the evaluation of QM energies and forces. How much computational speed up is then gained by the positional restraints applied to the MM atoms? Are the restraints enforced in all the simulations? Are the PBC/PME settings maintained in the restrained simulations?

Which are the details of the AM1/MM simulations? Why is AM1/MM used instead of MM? How are the initial structures selected? Which is the length of each AM1/MM simulation? Are positional restraints,

PBC, PME, ... applied?

For the sake of reproducibility and data availability, the MM parameters of the inhibitor molecules should be reported in separate files (e.g, mol2, frmod). Full coordinates of selected enzyme-inhibitor structures (maybe M062X/MM critical points and/or MD cluster representatives) would be required too.

- Finally, I find that the term “free energy surface (FES)” seems more adequate for 2D PMFs rather than for 1D PMD profiles.

Reviewer #2 (Remarks to the Author):

Several peptidyl inhibitors with the nitroalkene warhead were designed and synthesized. All compounds were tested in the FRET-based enzymatic assay and were found to have potent inhibition. FGA145, 146, and 147 showed antiviral activity in cell culture.

The X-ray crystal structures of FGA146 and FGA147 with Mpro were solved, revealing covalent inhibition and inhibitors induced conformational changes. The binding of inhibitor to Mpro was characterized by thermal shift assay and ITC. It was found that FGA145, FGA146, and FGA147 destabilize Mpro, while FGA159 stabilize Mpro. Both FGA145 and FGA146 showed tight binding to Mpro in the ITC assay. Overall, the exploration of nitroalkene as a reactive warhead for Mpro is significant, and the results are convincing and support the conclusion. The following comments need to be addressed:

1. "Recent situation in China shows the pandemic is far from over.

2"

Comment: I would suggest deletion of this sentence. It is OVER.

2. "Paxlovid, a combination of Mpro inhibitor nirmatrelvir and HIV protease inhibitor ritonavir"

Comment: this needs to be rephrased. Ritonavir acts as a metabolic booster in Paxlovid.

3. "New warheads to supplement the current repertoire would be welcome and potentially more effective."

Comment: Mpro inhibitors with new reactive warheads have been designed with potent enzymatic inhibition and antiviral activity. These key references should be cited:

Chem Sci. 2022 Feb 15;13(10):3027-3034.

J. Am. Chem. Soc. 2021, 143, 20697-20709.

ACS Med Chem Lett. 2022 Aug 11; 13(8): 1345–1350.

4. "CatL has been also recognized as a potential target for the search of drugs against COVID-19 as it is found to enable viral cell entry by activating the SARS-CoV-2 spike protein by cleavage.²⁶⁻²⁹

Comment: it should be acknowledged that dual inhibitors of Mpro and cathepsin L have been well documented in the literature. The key references are listed below:

Sci. Adv. 2020, 6, eabe0751.

ACS Infect Dis. 2021 Jun 11;7(6):1457-1468.

5. Table 1 needs to include a positive control such as nirmatrelvir or GC-376. It is also confusing that the authors used the Mpro expressed from two vectors pMal-Mpro and pET21-Mpro, which gave different K_i values. The authors need to explain why this is the case. The Mpro sequence should be the same regardless of the vector used, unless it is modified at either N or C terminus.

6. What is the positive control in Fig. 1? There is also no standard deviation in the EC50 plots. Importantly, it should be mentioned that Huh-7-ACE2 cells used is TMPRSS2 negative, meaning that SARS-CoV-2 enters cell mainly through endocytosis. Therefore, the observed antiviral effect is a combination of Mpro and cathepsin L inhibition. This is critical and needs to be discussed. To differentiate the inhibition of Mpro and cathepsin L, the authors might consider SARS-CoV-2 pseudovirus entry assay in different cell lines. However, additional experiments are not needed for this ms. ACS Infect. Dis. 2021, 7, 586-597.

7. Table 2. Since the authors argue that targeting cathepsin L might be beneficial in inhibiting SARS-CoV-2. It is suggested to change the off-targets to host proteases.

8. Since the X-ray crystal structures were solved, the computational analysis is less significant and should be moved to SI, unless it provides new information that was not revealed by the X-ray crystal structures.

9. The authors claimed that nitroalkene compounds are reversible covalent inhibitors of Mpro. What are the experimental evidences to support this claim? This can be demonstrated by enzymatic kinetic studies and fast dilution experiments.

Reviewer #3 (Remarks to the Author):

Competing Interests:

I work with the COVID-Moonshot team and the ASAP Discovery team on inhibitors of SARS-CoV2 main protease (the same target that is discussed here). I have no commercial interests in that our work is already funded and I have no shares or intellectual property interests in any SARS-CoV2 inhibitors.

The paper is well written and reasonably constructed. However, the central claim that the inhibitors FG145, FG146, and FG147 have promise to be drugs is not supported. Further the context of the compounds described in relation to critical existing literature for SARS-CoV2 MPro inhibitors needs to be given.

This area is now very well studied and to make the claim that these are potential drugs it would be necessary to demonstrate, with in vivo data, that there is the possibility of keeping blood levels of the free drug above the EC90 for cellular inhibition of SARS-CoV2 for 24h. As a minimum this could be done

by demonstrating pharmacokinetics in a rodent with measured human & rodent protein binding, microsome and hepatocyte clearance data. This could be used to generate a predicted human pharmacokinetic profile. In addition to claim to be potentially useful drug candidates initial toxicology profiling is needed. For covalent compounds I would expect to see a GSH inactivation rate, and also as nitro compounds are being proposed, minimally a 2 strain Ames test for mutagenicity. Protease activity is a particular concern as FG145 is described as a highly potent inhibitor of rhodesain, cruzain, cathepsin L and cathepsin B. Additional protease profiling is needed (for example the Nanosyn protease panel <http://nanosyn.bio/targets/protease/> which covers all the major protease classes) and an off target selectivity panel such as the CEREP 44 panel of critical off targets.

In this area demonstration of antiviral efficacy in a mouse SARS-CoV2 model would be desirable but is not essential if the pharmacokinetic profiling has been done.

Operation of these cellular viral assays is technically challenging therefore known literature compounds should be used and shown as standards for reference, ideally nirmatrelvir and ensitrelvir, but also the Pfizer and Groutas analogues (see below) PF-00835231 and GC376.

Finally, there is no reference to how the compounds were designed particularly with reference to the state of the literature when the work was conceived. This is essential as FGA147 is a direct nitroalkene analogue of the well established feline coronavirus protease inhibitor GC376 (Groutas 2013: <https://doi.org/10.1016/j.antiviral.2012.11.005>) and FGA146 is a direct analogue of the Pfizer human SARS Mpro inhibitor PF-00835231 (<https://pubs.acs.org/doi/10.1021/acs.jmedchem.0c01063>), which is the progenitor for the clinically used inhibitor nirmatrelvir. The publication that describes PF-00835231 also describes an extremely close acrylate ester (compound 2), which is another direct analogue of FGA147. Samples of PF-00835231 and GC376 should be obtained and the enzyme and cellular assays repeated with FGA146 and FGA147 to give accurate comparative data.

Comparison of the measured protein-ligand crystal structures should be presented in the context of the large number of SARS-CoV2 crystal structures available in the PDB and in particular that of GC376 (7CB7) and PF-00835231 (8DSU).

Castelló, September 21th, 2023,

TO WHOM IT CORRESPONDS

We take pleasure in enclosing a revised version of the manuscript entitled: “Peptidyl Nitroalkene Inhibitors of Main Protease (M^{Pro}) rationalized by Computational/Crystallographic Investigations as Antivirals against SARS-CoV-2”, to be considered for publication in *Communications Chemistry*. All the comments and criticisms raised by the reviewers have been discussed and/or addressed, as detailed below point by point.

As a general comment, we want to thank the reviewers, not only for their positive comments and global evaluation of our study, but for their suggestions that have been used to improve the quality of the manuscript. In particular, the comments, suggestions and questions raised by the reviewers have been addressed as follows:

Responses to Reviewer 1

Reviewer’s general comment: *“This article by Medrano et al. reports both experimental and computational results concerning the development of covalent inhibitors of the main protease (M^{Pro}) of SARS-CoV-2, which is a validated target for the treatment of the COVID-19 disease. Since protease inhibitors that have been approved as antiviral agents (e.g., nirmatrelvir) present various limitations, there is an ongoing interest in the development of new compounds as noticed by the authors. Basing on previous findings of dipeptidyl nitroalkenes as potent reversible inhibitors of cysteine proteases (ref. 23), this work demonstrates now that tailored peptidomimetic molecules equipped with the nitroalkene group form covalent adducts with the Cys145 thiol group at the active site of M^{Pro} and act as competitive in-vitro inhibitors of this enzyme. The best compounds (FGA145, FGA146 and FGA147) are characterized by inhibition constants K_I and dissociation constants K_D in the low micromolar range. Two crystallographic structures of M^{Pro} in complex with FGA146 and FGA147 are reported that provide valuable structural information. These results together with other data (cytotoxicity assays, binding isotherms and ITC, off-target inhibition studies) should support peptidyl nitroalkenes as promising candidates in future studies aimed to find potent (nM) M^{Pro} inhibitors. Nevertheless, it turns out also that further work is clearly necessary to elucidate the reversible (or irreversible) mode of action of the M^{Pro} inhibitors as well as to determine the actual significance of the computational results in comparison with the experimental data. More particularly, additional*

data and an improved description of the computational protocol are necessary to give sufficient methodological detail for the reproducibility of the simulations. Overall, I could recommend the publication of a revised version of the manuscript provided that the authors take into account and/or give reasonable answers to the following specific questions and comments:

Response: We thank the reviewer for his/her positive opinion on our study, despite their questions and concerns that helped us to improve the quality of the manuscript. Following his/her suggestions, additional experimental work to confirm the reversibility of the inhibitors has been carried out, also the section devoted to the description of the computational methods has been improved, including the estimation of the binding step through QM/MM alchemical calculations. Specific changes and additions are marked in the text and in the supporting information (files for review only), most of them detailed in the answers to the specific comments of the reviewer (below).

Reviewer's 1st comment: *"In the "Design and synthesis ..." subsection, the authors merely indicate that they designed six inhibitors. In the revised manuscript, they should describe the guidelines followed (or tools employed) in the molecular design of the inhibitors."*

Response: A new sentence has been added in page 4 to describe the design of inhibitors FGA145 and FGA146, including two new references: "The design of the new inhibitors was based upon nitroalkene inhibitors reported by us^{26,27} and previous Mpro inhibitors: compound **FGA146** is a direct analogue of the Pfizer human SARS M^{pro} inhibitor PF-00835231,²⁰ which is the progenitor for the clinically used inhibitor nirmatrelvir and **FGA147** is a direct nitroalkene analogue of the well-established feline coronavirus protease inhibitor GC376.²⁷"

Reviewer's 2nd comment: *"The authors claim in the abstract and in the Conclusions that their peptidyl nitroalkenes are reversible inhibitors of the Mpro enzyme. This reversibility is highlighted as important to reduce possible side effects associated to the inhibitors. However, in contrast with the approach followed in ref. 23, in the present work, there is no experimental verification of the reversibility of the inhibition via dialysis assays or dilution assays showing how fast the MPro activity is recovered. Such experimental proof should be incorporated to the revised manuscript."*

Response: Dilution assays to check for the reversibility of the binding of the inhibitor FGA146 to M^{pro} have been carried out. Reversible inhibitor nirmatrelvir and irreversible inhibitor LM188 were used as controls. The experiments confirm the reversibility of the nitroalkene inhibitor. A new section "Reversibility" has been introduced in page 20, including a new figure: "Figure 11. Reversibility of the binding of the inhibitor FGA146 to M^{pro}." "Reversibility. The results of the experiments carried out by dilution to check for the reversibility of the binding of the inhibitor FGA146 to M^{pro} is show in Figure 11. The activity of the MPro preincubated with FGA146 seems

to recover over time, which makes it likely that the compound is reversible. For the reversible control Nirmatrelvir there is only partial recovery of activity, and for the irreversible control there is no visible recovery of activity.” In addition, details of the dilution assays are explained in the Methods section in page 26.

Reviewer’s 3rd comment: *“The Methods section of the paper reproduces the Supporting Information (perhaps this is a formatting or uploading issue). A much shorter Methods section would be adequate for the main text. Note that there are two different Tables in the SI numbered as S2.”*

Response: We deeply acknowledge the comments of the reviewer. Following his/her suggestion, the Methods section has been shortened, and the details in the SI has been increased. However, we had to add two new paragraphs: one in page 26 to describe the reversibility assays and another one in page 26 to briefly describe the alchemical free energy calculations. An extended version with the reversibility assays and computational details is provided in the SI in pages S40-42 and S77. The typo error has been corrected; we thank the reviewer for the deep reading of our manuscript.

Reviewer’s 4th comment: *“Figure 10 displays two binding isotherms for compounds FGA145 and FGA146, and one ITC profile for FGA147. However, the details of the ITC experiments or ultracentrifugation are not included in the Methods section so that they must be added to the revised paper. It is unclear why the authors selected ultracentrifugation to analyze the binding of FGA145 and FGA146 to Mpro, while ITC is used to characterize the binding of FGA147. Hence, it would be highly convenient to compare the binding affinity of the three inhibitors using the same methodology, preferably ITC. Perhaps technical (solubility?) issues may complicate the ITC experiments, but the availability of the ITC-based free energies for FGA146-7 would be a valuable complement to the crystallographic structures.”*

Response: A detailed description of the experiments has been added in Methods section in page 25 (Binding experiments), as well as the justification for the use of centrifugation and ITC for the different inhibitors.

Reviewer’s 5th comment: *“Considering that the reported compounds inhibit the Mpro activity in the low micromolar range, the authors should most likely point out in the Discussion that further molecular design and optimization will be required to reach the nanomolar inhibition displayed by other inhibitors like nirmatrelvir. The usage of the adjective “potent” in the Abstract does not seem adequate.”*

Response: Discussion section and Abstract has been modified in agreement with the reviewer’s suggestions. Abstract has been modified in the new version and the term “potent” has been now

removed. Also, in page 17 the comment “and are potent inhibitors of its enzymatic activity” has been removed.

Reviewer’s 6th comment: *“The discussion of the computational results is quite confusing in general. For example, it is mentioned (p 19, l 408) that the activation of Cys145 by His41 takes place concertedly with the bond formation between the inhibitor and the enzyme, but the structural evidence supporting this claim is missing. Similarly, it is emphasized (p 19, l 406) that the “complete” free energy landscape of the inhibitor-enzyme covalent complex formation is generated, but the step of non-covalent binding ($E+I \rightarrow E:I$) is not investigated. It is stated that FGA146 and FGA147 are kinetic and thermodynamically “indistinguishable” in their binding to Mpro (p 19, l 412) and, in the same paragraph, that the different P3 fragment in the two compounds has a significant effect in their reactivity (p 19, l 419). The authors also consider that the predicted “thermodynamic behavior of FGA146 and FGA147 agrees with the very close experimentally determined K_i values”. However, the schematic free energy profile in Figure 2 shows that the Mpro-FGA146 adduct is 5.8 kcal/mol more stable than Mpro-FGA147, which corresponds to a 10000-fold difference in binding affinity! It is not clear either how the supposed reversibility of the inhibitors can be compatible with the relatively-large free energy barriers for the reverse process (25-31 kcal/mol; Figure 2). Overall, these and other observations (see below) suggest that the actual significance of the computational results is scarce”.*

Response: We thank the reviewer for his/her comments. Thus, following his/her suggestions, we have explored the non-covalent binding step ($E+I \rightarrow E:I$) by means of QM/MM alchemical free energy calculations. The results have been incorporated in the text and have been properly discussed. In fact, thanks to these new results, the agreement between the predicted thermodynamics and the measured inhibition equilibrium constants, is dramatically improved. Thus, the activation free energies that were already in very good agreement according to the similar kinetic behavior of the two studied inhibitors FGA146 and FGA147, are now complemented with the very close reaction free energies measured from the solvent separated $E + I$. Regarding the discussion on the computational results, the question on whether the activation of Cys145 by His41 takes place concertedly with the bond formation between the inhibitor and the enzyme, which was justified by the free energy profiles (previous and new Figure 2) and the structures of $E:I$ and $E-I^{(-)}$ (previous Figure 4), is now supported by information and detail of the structures of TS1 and TS2 (see new Figure 3, Table S7 and S8). Finally, we acknowledge the reviewer for pointing out the error when discussing the effect of P3 in the kinetics, which is indeed irrelevant. In the manuscript a new Scheme 2, a new Figure 2, a new Figure 3, and a new Figure 4 have been added. Also new text has been added in the manuscript in pages 12,13 and 21,22. New text has been added in the Supporting Information in page S4S40-S42, and new Tables S7, S8, S9 and S10.

Reviewer's 7th comment: *"The presentation of the energetic and structural data produced by the simulations should be improved to provide a clear description of the reaction mechanism. For example, the schematic free energy profiles in Figure 2 should be replaced by (or at least augmented with) the corresponding PMF plots including the data points associated to each window (i.e., not just the interpolation curve). This energetic information should be complemented with molecular models showing the most relevant features of the intermediate and TS-like configurations as well as with the corresponding plots showing the evolution of the reactive distances along the reaction coordinate s (I note in passing that a proper definition of s within the space expanded by the reactive distances is missing). In the supplementary material, Table S7 collects key inter-atomic distances for the initial E:I and the final E-I states, but other structures along the reaction coordinate like the E-I(-) intermediate and the TS-configurations should be included, identifying also the origin of such distances (average values from M062X/MM PMF windows? M062X/MM optimized structures?). Table S3 and S7 (including interatomic distances between heavy atoms) may be merged to facilitate the comparison between X-ray and computational models."*

Response: We thank the reviewer for his/her comments. Following his/her suggestions, the new Figure 2, that includes the binding step in the overall free energy profile, is now complemented, as well as with the computed PMFs plots as interpolated curve but also with the data points associated with each window (see new Figure S11 and S12). We think that keeping the profile in the main text as a figure (Fig. 2) can help the reader to follow the discussion of the text. Models of the corresponding states, including E-I⁽⁻⁾, TS1 and TS2, are shown now in Figure 4. We appreciate the suggestion of merging Table S3 with Table S7 (and Table S8). However, considering that the distances of the X-ray structures listed in Table S3 are limited to the E-I complex and Tables S7 and S8 includes all states along the reaction, we prefer to keep the information as separated tables. We address the reader to compare both tables in the text (page 16-17). Moreover, the overlay of structures derived from X-ray and calculations (Figure 7) can also help in understanding the discussion about the very good agreement. New text has been added in the manuscript in pages 17-18. The new Figure 4 has been also added. New text appear now in the Supporting Information in page S41, Table S7, S8, and Tables S9 and S10.

Reviewer's 8th comment: *"The crystallographic 8BGA and 8BGD structures display the flexibility of several structural motifs, which delineate the active site region (P2-helix in the S2 subsite, P5 loop, P4 flap,...) This is a particularly relevant observation that is in line with former assessments of other X-ray structures of MPro (e.g., ref 33). Consequently, the ability of the MD simulations to shed light onto this flexibility and/or to compare with the X-ray models will require*

the sampling of the slow backbone motions over hundreds of nanoseconds. In the present work, the X-ray models of the E-I complexes are just compared with a “single” QM/MM MD structure (Figure 7). Although the settings of the QM/MM MD calculations (presumably, AM1/MM MD) remain essentially unknown (see below), the length of such simulations seems likely to be below 2-5 nanoseconds given that only 1000 MD frames are used for averaging the interaction energies (Figure 3). Hence, two important problems with the simulations arise at this point: (i) the enzyme loop motions are undersampled; (ii) there is no proper analysis of the MD data in order to select and characterize statistically-sound representative structures to be compared with the X-models. Sufficiently-extensive and unconstrained simulations in explicit solvent using classical force fields followed by well-designed clustering analysis, RMSD calculations, ... may be helpful.”

Response: We have done our best to expand and improve the description of the methods, following the reviewer’s suggestion.

Regarding the discussion about the dynamics, it is important to point out that the interaction between the inhibitor and the protein are based on AM1/MM MD simulations, not on classical MD simulations. Consequently, this interaction energies are computed at QM level, despite this decision force to use shorter MD trajectories. Because no restrains were applied to the inhibitor or the protein, the flexibility of the systems are preserved. The computationally predicted structures displayed in Figure 7 are representative of the AM1/MM MD simulations while the overlap of 100 frames of the two inhibitors in the active site, computed along 100 ns of classical MD simulations, is displayed in Figure 8. At this point, we must point out that the enzyme loops are free to move during the simulations. We only display a single structure of the protein in Figure 8 for clarity purposes. We have improved the description of the captions in the new version of the manuscript.

We agree with the reviewer that other approaches can be adopted, based on extensive exploration of the conformational space with classical force fields, but we consider our approach is the most appropriate to study the chemical reactivity of the inhibition process with, as shown, excellent agreement between the structures obtained from the calculations and the X-ray diffraction studies. New text has been added in the manuscript in pages 16-17, new caption of Figure 7 and 8, and new text in the Supplementary Material.

Reviewer’s 9th comment: *“Figures S13 and S14 can be removed because they do not add any relevant information with respect to Figure 3. In addition, the legend of Figure 3 needs to be improved by including the description of items a), b), c), and d). In the legend of Figure S12, include the name of the other inhibitors, the corresponding PDB codes and the criteria used for the superposition of the different structures. The interaction between the phenyl group in P3 of FGA147 with Gln189 and Glu166 (p 16, I 336) that explains its orientation is not shown in Figure*

3. Again, clustering calculations are required here to characterize quantitatively the conformations adopted by the P3 residue in FGA146 and FGA147. A representation of the most populated clusters should be shown in Fig. 8 while the statistical abundance of the inhibitor...oxyanion hole contacts should be reported.”

Response: Following reviewer's suggestion, Figures S13 and S14 has been removed. We considered they provided information of all the residues, but it is true that it was not additional relevant information. Also, Figure 3 (now Figure 4) has been improved, same as its caption, following the reviewer's suggestion. The caption of Figure S13 (S12 in previous version) has been improved. The interaction between P3 and Gln189 is shown as an orange bar in Figure 4. It is difficult to show figures that clearly display all the mentioned interactions, which is the reason we consider that the quantitative QM/MM analysis summarized in Figure 4 is adequate and provides very valuable information. We decided to show the superposition of the structures generated during the MD simulations in Figure 8, which gives a clear indication of the flexibility of the two inhibitors in the active site of the enzyme. New text has been added in the manuscript in page 10, together with new Figure 4 and its caption. New text appears also in the Supporting Information in page S51, and new caption of Figure S13.

Reviewer's 10th comment: *“The Comm Chem guidelines for referees remark the importance that authors provide enough data or methodological information to help others replicate their work. Although the reproducibility of complex simulations is not trivial, the lack of technical details/settings and relevant structures in the supporting material make things worse. For example:*

What QM level of theory is used to obtain the RESP charges (AM1-bcc, HF/631G,...)? Which version of GAFF is used?*

Which is the minimum distance between solute atoms and the solvent box walls?

How do the RMSD plots look like?

How are the end structures for the NEB calculations selected or built? Which are the NEB settings (number of images, spring constants, ...)?

How many critical points are located on the static reaction paths at the M062X/MM level? Which are their properties and geometries? How many points are obtained in the MEP calculations?

How many windows are considered in the PMF calculations? Is there an equilibration period within each window? Which criteria are followed to assess the convergence of the PMF profiles with the total simulation time?

What is the exact definition of the collective variable adopted for the PMFs?

The computational bottleneck of QM/MM simulations is usually due to the evaluation of QM energies and forces. How much computational speed up is then gained by the positional

restraints applied to the MM atoms? Are the restraints enforced in all the simulations? Are the PBC/PME settings maintained in the restrained simulations?

Which are the details of the AM1/MM simulations? Why is AM1/MM used instead of MM? How are the initial structures selected? Which is the length of each AM1/MM simulation? Are positional restraints, PBC, PME, ... applied?

For the sake of reproducibility and data availability, the MM parameters of the inhibitor molecules should be reported in separate files (e.g. mol2, frcmod). Full coordinates of selected enzyme-inhibitor structures (maybe M062X/MM critical points and/or MD cluster representatives) would be required too.”

Response: We acknowledge the reviewer's suggestions. We agree with him/her that any study must report enough information to replicate the reported data. However, the amount of data that is generated by a computational study such as the one we are presenting here is gigantic, which prevents reporting all them. The information we are reporting is not much different to the information reported in related studies. As stated by the reviewer, the reproducibility of complex simulations is not trivial. Anyway, following his/her suggestions, we have improved the description of all the technical details/settings required by the reviewer, as follows:

- The RESP charges derived with Antechamber are based on AM1-BCC, as stated in the captions of Tables S4 and S5. We compared those partial charges with the corresponding ones derived using the HF/6-31(g) based on a previous M06-2x/6-31+g(d,p) optimization, and the ones provided from AM1-BCC render softer values (mostly in some aliphatic sp^3 carbons). This is the reason we chose the AM1-BCC charges instead (as also recommended in the Amber manual).
- We used the default GAFF version.
- The *solvateBox* command of the *tLeap* facility was used with a value of 10 Å for the *distance* and 0.8 Å for the *closeness*. We have included this information in the Computational Methods of the Supplementary Material.
- The RMSD plots have are shown in the Figure S10 of the Supplementary Material. Both present plateau regions suggesting that partial thermal equilibrium has reached.
- Each NEB was constructed using a total of 60 geometries initially obtained from linear interpolation between the optimized E:I, E-I⁽⁻⁾ and E-I structures. Then umbrella harmonic restraints with a force constant of 200 kJ/mol.Å² was applied between previous and following geometries. Further search saddle points were carried out from the maxima found in the resulting NEB optimized geometries.
- Two TSs in the FGA146 inhibitor, corresponding to the initial proton transfer Cys-His (729.6i cm⁻¹), and the one associated to the transfer His-C20 (1227.6i cm⁻¹), and two TSs in the FGA147, corresponding to the initial proton transfer Cys-His (1191.8i cm⁻¹),

and the one associated to the transfer His-C20 ($1188.4i \text{ cm}^{-1}$), were located and optimized as saddle point of order one at DFT/MM level.

- PMFs were based on the path collective variable s , thus we used the geometries obtained during the PES exploration as milestones. As a consequence, a total of 60 windows by PMF were used (during the WHAM integration a few windows were destined, based on the overlapping among them). The definition of the collective variable has been incorporated to the Computational Methods section of the Supplementary Material. Regarding the MD at each window of the PMF, since they are based on M06/MM potentials, and we are using a small time step (0.5 fs, due the presence of proton transfers), there's an obvious limitation on the amount of time we can accumulate. In these cases, only a few initial data is not considered in the integration (100 – 400 initial steps), which usually corresponds to the initial thermal equilibration (Langevin thermostat is a very efficient one, just needing a few steps to reach the desired temperature). The WHAM convergence was set to 10^{-3} in the maximum free energy difference between iterations, and the results are also compared with the ones obtained via Umbrella Integration.
- Our standard protocol for QM/MM calculations usually keeps frozen the large part of the model lying out of a spherical cutoff around QM atoms (indeed we usually keep fixed any atom not included within the electronic QM-MM charge interaction integrals). Whereas for semiempirical methods this can be useful, in this case where DFT/MM calculations are being used maybe it does not result in an appreciable reduction of CPU time. In any case, considering that the QM implementation uses a truncation scheme within the mono-electronic interactions integrals, keeping the outer part frozen may avoid diffusion problems (mostly with the solvent molecules).
- We have used the same non-bonding conditions (PBC/PME) along all the simulations.
- Regarding the AM1/MM simulations, we used the same conditions as in the DFT/MM ones but switching to a semiempirical method. The motivation is being able to run longer times than in the DFT method, and thus obtain reliable averages. We performed 100 ps for calculating the interactions energies, and the dynamics performed on the free energy maxima were restrained using a tether with a cartesian force constant of 3500 kJ/mol.Å^2 .
- As stated in the manuscript ("Data availability", page 27) any additional data can be provided by the authors upon request.

New text has been added in the manuscript in page 26-27 (QM/MM Simulations section) and in the Supporting Information in pages S39-S41 and Figures S10, S12 and S13, and Tables S4, S5, S9 and S10.

Reviewer's 11th comment: *"Finally, I find that the term "free energy surface (FES)" seems more adequate for 2D PMFs rather than for 1D PMD profiles"*

Response: We respectfully disagree with the reviewer in this point. FES is a term that defines a N-dimensional function, whether this is projected into two or one dimensions.

Responses to Reviewer 2

Reviewer's general comment: *"Several peptidyl inhibitors with the nitroalkene warhead were designed and synthesized. All compounds were tested in the FRET-based enzymatic assay and were found to have potent inhibition. FGA145, 146, and 147 showed antiviral activity in cell culture. The X-ray crystal structures of FGA146 and FGA147 with Mpro were solved, revealing covalent inhibition and inhibitors induced conformational changes. The binding of inhibitor to Mpro was characterized by thermal shift assay and ITC. It was found that FGA145, FGA146, and FGA147 destabilize Mpro, while FGA159 stabilize Mpro. Both FGA145 and FGA146 showed tight binding to Mpro in the ITC assay. Overall, the exploration of nitroalkene as a reactive warhead for Mpro is significant, and the results are convincing and support the conclusion."*

Response: We thank the reviewer for his/her positive opinion on our study, despite their questions and concerns that helped us to improve the quality of the manuscript. Following his/her suggestions, the section devoted to the description of the computational methods has been improved, as well as additional work to confirm the reversible character of the mode of action of the selected inhibitors. Specific changes and additions are detailed below.

Reviewer's 1st comment: *"Recent situation in China shows the pandemic is far from over. 2"*

Response: Following the reviewer's suggestions, the sentence has been removed.

Reviewer's 2nd comment: *"Paxlovid,7 a combination of Mpro inhibitor nirmatrelvir and HIV protease inhibitor ritonavir"*

Response: Following the reviewer's suggestions, the text has been modified. In the Introduction in page 1 now says: "Among approved antivirals for the treatment of COVID-19 are nucleoside derivatives remdesivir⁴ and molnupiravir,⁵ with uncertain efficacy for certain types of patients, and Paxlovid,⁶ a combination of the protease inhibitor nirmatrelvir and ritonavir, a metabolic booster that increase the effectiveness of the protease."

Reviewer's 3rd comment: "New warheads to supplement the current repertoire would be welcome and potentially more effective."

Comment: Mpro inhibitors with new reactive warheads have been designed with potent enzymatic inhibition and antiviral activity. These key references should be cited:

Chem Sci. 2022 Feb 15;13(10):3027-3034.

J. Am. Chem. Soc. 2021, 143, 20697-20709.

ACS Med Chem Lett. 2022 Aug 11; 13(8): 1345–1350."

Response: Following the reviewer's suggestions, the text has been modified. The sentence mentioned by the reviewer appearing in page 2 has been replaced by a new one: "M^{pro} inhibitors with new reactive warheads, a-chlorofluoroacetamide², dichloroacetamide³, and nirmatrelvir analogs⁴, have been designed with potent enzymatic inhibition and antiviral activity." Including the references suggested by the reviewer.

Reviewer's 4th comment: "CatL has been also recognized as a potential target for the search of drugs against COVID-19 as it is found to enable viral cell entry by activating the SARS-CoV-2 spike protein by cleavage.^{26-29'} *Comment: it should be acknowledged that dual inhibitors of Mpro and cathepsin L have been well documented in the literature. The key references are listed below:*

Sci. Adv. 2020, 6, eabe0751.

ACS Infect Dis. 2021 Jun 11;7(6):1457-1468."

Response: Following the reviewer's suggestions, the text has been modified. The new sentence: "Dual inhibition against SARS-CoV-2 M^{pro} and human Cathepsin-L have been reported.^{5,6}" has been added in page 2 including the two references suggested by the reviewer.

Reviewer's 5th comment: "Table 1 needs to include a positive control such as nirmatrelvir or GC-376. it is also confusing that the authors used the Mpro expressed from two vectors pMal-Mpro and pET21-Mpro, which gave different Ki values. The authors need to explain why this is the case. The Mpro sequence should be the same regardless of the vector used, unless it is modified at either N or C terminus"

Response: A control with nirmatrelvir has been added to table 1. The differences might come from the different amount of organic solvent used in the experiments, 1% versus 7.5%. This has been added to the the manuscript in page 22 in Methods (Enzymatic assays). Organic solvents are known to affect the stability and the enzymatic activity of proteins and represent an important factor when designing experiments to develop new drugs. See references:

Griebenow, K.; Klibanov, A.M. On Protein Denaturation in Aqueous-Organic Mixtures but Not in Pure Organic Solvents. *J. Am. Chem. Soc.* **1996**, 118, 11695–11700.

Kamal, M.Z.; Yedavalli, P.; Deshmukh, M.V.; Rao, N.M. Lipase in aqueous-polar organic solvents: Activity, structure, and stability: Lipase in Organic Solvents. *Protein Sci.* **2013**, *22*, 904–915.

Jacobson, A.L.; Turner, C.L. Specific solvent effects on the thermal denaturation of ribonuclease. Effect of dimethyl sulfoxide and p-dioxane on thermodynamics of denaturation. *Biochemistry* **1980**, *19*, 4534–4538.

Magsumov, T., Fatkhutdinova, A., Mukhametzyanov, T. and Sedov, I. The Effect of Dimethyl Sulfoxide on the Lysozyme Unfolding Kinetics, Thermodynamics, and Mechanism. *Biomolecules* **2019**, *9*, 547.

The effect of DMSO and DMF (N,N-dimethylformamide) was tested on the enzymatic activity of M^{pro} (see figure). There was a clear effect of these organic solvents on the activity of the protein. Ethanol did precipitate the protein at 7.5%.

Reviewer's 6th comment: "What is the positive control in Fig. 1? There is also no standard deviation in the EC50 plots. Importantly, it should be mentioned that Huh-7-ACE2 cells used is TMPRSS2 negative, meaning that SARS-CoV-2 enters cell mainly through endocytosis. Therefore, the observed antiviral effect is a combination of M^{pro} and cathepsin L inhibition. This is critical and needs to be discussed. To differentiate the inhibition of M^{pro} and cathepsin L, the authors might consider SARS-CoV-2 pseudovirus entry assay in different cell lines. However, additional experiments are not needed for this ms.

ACS Infect. Dis. 2021, *7*, 586-597."

Response: We fully agree with the reviewer that in TMPRSS2-negative cell lines, SARS-CoV-2 uses instead of TMPRSS2 a CatL-dependent viral entry route. Therefore the combination as pointed out, is likely and moreover desired. So far, to the best of our knowledge, the real expression levels of Huh-7 cells (in contrast to conventionally used Vero E6 cells) regarding TMPRSS2 is rather unknown. There is only one publication claiming the expression in Huh-7 cells ([https://www.cell.com/iscience/pdf/S2589-0042\(21\)00388-6.pdf](https://www.cell.com/iscience/pdf/S2589-0042(21)00388-6.pdf)). Therefore, we have added the following sentence in page 6: "To validate the therapeutic potential of the designed compounds, antiviral assay in suitable, primary cell systems (e.g. human bronchial epithelial cells) needs to be done in the future."

We moreover decline to show SDs for the EC50-curves due to high variability in (relative) percentage values when high viral titers are achieved. This is due to the experimental setup of plaque assays. Since here plaques per well were counted after a 10-fold dilution series, samples with high viral titers show higher variations when compared in relation to the untreated control. To increase transparency, we showed virus titer reductions of at least three independent experiments on the (absolute) virus titer levels (indicated as dot per experiment).

Reviewer's 7th comment: *"Table 2. Since the authors argue that targeting cathepsin L might be beneficial in inhibiting SARS- CoV-2. It is suggested to change the off-targets to host proteases"*

Response: The term has been changed as suggested.

Reviewer's 8th comment: *"Since the X-ray crystal structures were solved, the computational analysis is less significant and should be moved to SI, unless it provides new information that was not revealed by the X-ray crystal structures"*

Response: With all our respects, we deeply disagree with the reviewer. The computational results provide information about the mechanistic details of the inhibition process at atomistic level. This information refers not only to the final state (the covalent E-I complex) where X-ray studies provide geometrical information, but along all states of the reaction. This information can be crucial in order to support the improvement of the inhibitors by a rational design, that can be based on the reduction of activation free energies, the improvement of the binding process, etc. Moreover, the results of the simulations presented in the present manuscript provides dynamic information of the geometries as well as allowing to carry out a quantitative analysis of interactions, not only deduced from close contact interactions derived from X-ray structures. However in the new version of the manuscript, computational calculations section have been moved after the crystallographic studies (in the Results and in the Discussion sections).

Reviewer's 8th comment: *"The authors claimed that nitroalkene compounds are reversible covalent inhibitors of Mpro. What are the experimental evidences to support this claim? This can be demonstrated by enzymatic kinetic studies and fast dilution experiments."*

Response: Dilution assays to check for the reversibility of the binding of the inhibitor FGA146 to M^{pro} have been carried out. Reversible inhibitor nirmatrelvir and irreversible inhibitor LM188 were used as controls. The experiments confirm the reversibility of the nitroalkene inhibitor. A new section "Reversibility" has been introduced in page 20, including a new figure: "Figure 11. Reversibility of the binding of the inhibitor FGA146 to M^{pro}." "Reversibility. The results of the

experiments carried out by dilution to check for the reversibility of the binding of the inhibitor FGA146 to M^{Pro} is shown in Figure 11. The activity of the MPro preincubated with FGA146 seems to recover over time, which makes it likely that the compound is reversible. For the reversible control Nirmatrelvir there is only partial recovery of activity, and for the irreversible control there is no visible recovery of activity.” In addition, details of the dilution assays are explained in the Methods section in page 26.

Responses to Reviewer 3

Reviewer's general comment: *"The paper is well written and reasonably constructed. However, the central claim that the inhibitors FG145, FG146, and FG147 have promise to be drugs is not supported. Further the context of the compounds described in relation to critical existing literature for SARS-CoV2 MPro inhibitors needs to be given."*

Response: We thank the reviewer for his/her positive opinion on our study, despite their questions and concerns that helped us to improve the quality of the manuscript. Following his/her suggestions additional work to measure the pharmacokinetic properties of the selected inhibitors has been carried out. Specific changes and additions are detailed below.

Reviewer's 1st comment: *"This area is now very well studied and to make the claim that these are potential drugs it would be necessary to demonstrate, with in vivo data, that there is the possibility of keeping blood levels of the free drug above the EC90 for cellular inhibition of SARS-CoV2 for 24h. As a minimum this could be done by demonstrating pharmacokinetics in a rodent with measured human & rodent protein binding, microsome and hepatocyte clearance data. This could be used to generate a predicted human pharmacokinetic profile. In addition to claim to be potentially useful drug candidates initial toxicology profiling is needed. For covalent compounds I would expect to see a GSH inactivation rate, and also as nitro compounds are being proposed, minimally a 2 strain Ames test for mutagenicity. Protease activity is a particular concern as FG145 is described as a highly potent inhibitor of rhodesain, cruzain, cathepsin L and cathepsin B. Additional protease profiling is needed (for example the Nanosyn protease panel <http://nanosyn.bio/targets/protease/> which covers all the major protease classes) and an off target selectivity panel such as the CEREP 44 panel of critical off targets.."*

Response: Following the reviewer's suggestions, a physicochemical study for best inhibitor FGA146 has been carried out. The calculated physicochemical assays include: calculation of physicochemical parameters, stability, general electrophilic reactivity (cysteine assay), logP determination, permeability, reversibility, selectivity against serine proteases, selectivity against another target with cysteine, cytotoxicity and off-target prediction. The results of all the assays have been included in the Supplementary material, and new text and figure about the reversibility assays have been included in the manuscript: in Results in page 20 including Figure 11 and in Methods page 25.

Reviewer's 2nd comment: "In this area demonstration of antiviral efficacy in a mouse SARS-CoV2 model would be desirable but is not essential if the pharmacokinetic profiling has been done."

Response: We agree on in vivo data regarding the compounds. Since rodent models need intensive planning and regulations, the data cannot be generated in a short period of time. Pharmacokinetic profiling has been performed as explained above.

Reviewer's 3rd comment: "Operation of these cellular viral assays is technically challenging therefore known literature compounds should be used and shown as standards for reference, ideally nirmatrelvir and ensitrelvir, but also the Pfizer and Groutas analogues (see below) PF-00835231 and GC376."

Response: We have published data in the same cell system and infection parameters with SARS-CoV-2 on nirmatrelvir, which can be found here:

<https://www.sciencedirect.com/science/article/pii/S0223523422009230?via%3Dihub>

Compounds	EWG	AA	Ki (μM) or % of inhibition at 20 μM			
			SARS-CoV-2 M ^{pro}	hCatL	hCatB	
SPR35		Nva	1.77 ± 0.16	-	-	
SPR36		Tle	33 ± 2%	-	-	
SPR37		Leu	0.386 ± 0.055	-	-	
SPR38		Nle	0.260 ± 0.066	1.92 ± 0.10	11.1 ± 1.2	
SPR39		Cpa	0.252 ± 0.028	3.38 ± 0.20	7.88 ± 0.65	
SPR40		Tba	0.416 ± 0.058	-	-	
SPR41		Cha	0.184 ± 0.025	0.252 ± 0.018	14.4 ± 1.2	
SPR42		Nle	29 ± 4%	-	-	
SPR43		Tba	50 ± 5%	-	-	
SPR44		Cha	47 ± 2%	-	-	
Nirmatrelvir		-	-	0.003 ± 0.0004	-	-
11a		-	-	0.008 ± 0.0009	-	-

Table 2

Antiviral and cytotoxicity evaluation of the most promising SARS-CoV-2 M^{pro} inhibitors.

Cmpd	Huh-7-ACE2 cells infected EC ₅₀ (μM)	Huh-7-ACE2 cells CC ₅₀ (μM)	SI
SPR38	18.5 ± 6.5	60.9 ± 11.5	3
SPR39	1.5 ± 0.3	100	66.6
SPR41	1.8 ± 0.1	14.5 ± 3.4	8
Nirmatrelvir	<0.01	>100	>10.000

Reviewer's 4th comment: "Finally, there is no reference to how the compounds were designed particularly with reference to the state of the literature when the work was conceived. This is essential as FGA147 is a direct nitroalkene analogue of the well established feline coronavirus protease inhibitor GC376 (Groutas 2013: <https://doi.org/10.1016/j.antiviral.2012.11.005>) and FGA146 is a direct analogue of the Pfizer human SARS Mpro inhibitor PF-00835231 (<https://pubs.acs.org/doi/10.1021/acs.jmedchem.0c01063>), which is the progenitor for the clinically used inhibitor nirmatrelvir. The publication that describes PF-00835231 also describes an extremely close acrylate ester (compound 2), which is another direct analogue of FGA147.

Samples of PF-00835231 and GC376 should be obtained and the enzyme and cellular assays repeated with FGA146 and FGA147 to give accurate comparative data.

Comparison of the measured protein-ligand crystal structures should be presented in the context of the large number of SARS-CoV2 crystal structures available in the PDB and in particular that of GC376 (7CB7) and PF-00835231 (8DSU)..”

Response: We thank the reviewer for his/her suggestions.

A new sentence has been added in page 4 to describe the design of inhibitors FGA145 and FGA146, including two new references: “The design of the new inhibitors was based upon nitroalkene inhibitors reported by us^{26,27} and previous Mpro inhibitors: compound **FGA146** is a direct analogue of the Pfizer human SARS M^{pro} inhibitor PF-00835231,²⁰ which is the progenitor for the clinically used inhibitor nirmatrelvir and **FGA147** is a direct nitroalkene analogue of the well-established feline coronavirus protease inhibitor GC376.²⁷”

Regarding to the comparison of protein-ligand crystal structures, a new sentence has been also added in Discussion in page 21: “The crystal structures of M^{pro} in complex with **FGA146** and **FGA147** show a very similar binding mode to that of the similar inhibitors PF-00835231 and GC376. The only significant difference is the warhead and its binding to the “oxyanion hole” (see new **Figure S9**).”

Yours sincerely,

Florenci V. González

Reviewers' comments:

Reviewer #1 (Remarks to the Author):

In their revised manuscript, Medrano et al. have considered some of the changes requested by the reviewers and certainly improved the presentation of their results in the manuscript and in the SI. They have also fixed the mistakes detected by the three referees. However, after the long revision process, I think that there are still unsolved issues in this paper.

The reversibility of the inhibition by FGA146 has been examined by means of dilution assays (Fig. 11). But the interpretation of these experiments seems unclear because as noticed by the authors “the activity of the Mpro preincubated with FGA146 seems to recover over time, which makes it likely that the compound inhibits its target reversibly” (my italics). As a matter of fact, the authors have removed the expression “reversible inhibitors” from the abstract and the final summary. This seems at odds with the following statement from the rebuttal letter “work to confirm the reversibility of the inhibitors has been carried out.” Probably, further study would have been required to clarify this important point (why FGA145 and/or FGA147 were not considered for the dilution experiments?), but I believe that there is only weak evidence supporting the reversibility of the proposed inhibitors. In any case, a more clear discussion about this point seems necessary.

In the revised manuscript, the authors have explored the non-covalent binding step by means of AM1/MM alchemical free energy calculations. In their letter, they claim that “the agreement between the predicted thermodynamics and the measured inhibition equilibrium constants, is dramatically improved”. Thus, the combination of the binding free energy for the E+I→E:I process with the free energy for the chemical E:I→E-I step results now in total ΔG values of -28.9 and -31.7 kcal/mol for FGA146 and FGA147, respectively. It is simply stated in p 13 | 270-271 that “These results are in agreement with the almost equivalent experimentally measured equilibrium KI values” (i.e., around 1-2 μM). However, the calculated energies can be translated into effective KI values of 10^{-16} and 10^{-18} μM for FGA146 and FGA147. Apart from the large overestimation of the binding free energy (which raises serious concerns about the adequacy of the QM/MM methods and protocols), it turns out that FGA147 is predicted to be 100-fold more effective than FGA146. Maybe such difference could have been considered to be small, perhaps being comparable with the (undetermined) statistical uncertainty of the free energy calculations and/or in consonance within previous benchmarks of the theoretical methods against validated experimental targets, but no further information or explanation are provided. Therefore, the claim made by the authors about the agreement between experimental and computational data is unsupported.

In their letter (but not in the SI), the authors indicate that they performed 100 ps AM1/MM simulations for calculating the interaction energies (IEs) reported in Figure 6 for the E:I and E-I states. In view of the free energy profiles in Figure 4, the significance of the IEs is very limited. For example, the large difference in the stability of the E:I complexes (8.9 kcal/mol in favor of FGA146), cannot be rationalized in terms of the IEs: FGA147 has larger IEs with P39, D48, H49, N142, M165, H172, Q189, S1 and R4 while

only N28, E166 and R186 favor FGA146.

The same 100-ps AM1/MM MDs were used to obtain the QM/MM snapshots shown in Figures 7 and 8. However, it is well known that 100 ps MD trajectories do not provide significant sampling of the side chain and loop motions so that the comparison with X-ray data is statistically unreliable. Similarly, the relevance of the qualitative flexibility analysis in Figure 8 is very low. Therefore, the structural results from the AM1/MM simulations cannot be accepted for publication. Alternatively, the authors may focus on the comparison of key interatomic distances between the M062X/MM critical structures (E-I in Table S7) and X-ray data. In this respect, I note that TS2 configurations in Figure 5 are missing and they must be included. The relative M062X/MM energies of the optimized structures should be also reported in Figure 5 to assess the weight of thermal effects in comparison with the profiles in Figure 4.

Many more details of the computational protocol have been added to the revised SI (further revision of the English and typo corrections are required; there is even a hyphenated phrase in p. S40).

Nevertheless, there are a few points that are still unclear. For example:

- Prior to the NEB calculations, it seems that the authors carried out M062X/MM geometry optimizations of the E:I, E-I(-) and E-I configurations and then performed the corresponding NEB interpolations (BTW: this information is missing in the SI). How did the authors select the starting geometries for these calculations? Did they pick up representative MD snapshots according to well-defined criteria? This is indeed a critical point in the QM/MM protocol.
- In the response letter, it is noticed (i) "... interaction energies are computed at QM level, despite this decision force to use shorter MD trajectories. Because no restrains were applied to the inhibitor or the protein, the flexibility of the systems are preserved.", (ii) "Regarding the AM1/MM simulations, we used the same conditions as in the DFT/MM, ones but switching to a semiempirical method", and (iii) "Our standard protocol for QM/MM calculations usually keeps frozen the large part of the model lying out of a spherical cutoff around QM atoms". These seem to be rather contradictory statements. It is not clear then whether or not positional restraints were used in the AM1/MM MD simulations. I also note in passing that IEs are AM1/MM values, not just AM1.
- The authors note that "We have used the same non-bonding conditions (PBC/PME) along all the simulations". The PME settings for the MM MD calculations are standard, but according to the reference paper of the QMCube driver, the QM atoms are described as zero-charge vdW spheres in the MM engines. How do the authors treat the electrostatic QM...QM interactions across images?

In their letter, the authors remind me that "additional data can be provided by the authors upon request". Please note that, as a referee, I already did such request and I do it again here. In my first report, I pointed out that Comm Chem guidelines remark the importance that authors provide enough data or methodological information to help others replicate their work. Of course, this guideline is fully in consonance with the increasing importance of data and method sharing in the field of molecular modelling (<https://doi.org/10.1021/acs.jcim.3c00599> ; <https://doi.org/10.1021/acs.jcim.0c01389>). Thus, I would like to recommend the authors to see how a recent Comm Chem paper makes available datasets that contain scripts, coordinate files, solvent-stripped trajectories, etc. (<https://doi.org/10.1038/s42004-023-01006-0>). Hopefully, they may realize that their statement, "the amount of data that is generated by

a computational study such as the one we are presenting here is gigantic, which prevents reporting all them”, is far from being reasonable.

Reviewer #2 (Remarks to the Author):

Comments from the previous round of review were properly addressed. I therefore recommend acceptance.

Reviewer #3 (Remarks to the Author):

Most of the revisions are fine, however they have kept the statement

"All these results combined suggest the viability of employing these compounds as promising drugs against SARS-CoV-2 and new coronavirus that might appear in the future."

Given the poor cellular activity of the compounds (>1 μ M) and the complete lack of measured ADMET data, I can't agree with that statement.

These are interesting ligands, but there is no evidence presented that they are or can be drugs in the future.

The authors make the statement in the rebuttal letter:

"Response: We agree on in vivo data regarding the compounds. Since rodent models need intensive planning and regulations, the data cannot be generated in a short period of time. Pharmacokinetic profiling has been performed as explained above."

In vivo pharmacokinetic data has not been obtained, they haven't done the study needed to demonstrated free exposure over the cellular EC90. The statement made "Since rodent models need intensive planning and regulations, the data cannot be generated in a short period of time." Is factually untrue. There are numerous CROs who will conduct these studies on supply of material, it is a funding and contracting issue. I appreciate they may need to fund the study and supply appropriate material - but it is simply inadequate to make this excuse if they want to make that assertion.

The literature is plagued with ligands claimed to be drugs with no evidence of exposure, this creates a false hope and vastly underestimates the scientific challenges of actually generating new therapeutic agents.

Either they need to generate the data to support the assertion "All these results combined suggest the viability of employing these compounds as promising drugs against SARS-CoV-2 and new coronavirus that might appear in the future." or withdraw it.

Castelló, November 17th, 2023

Dear Sir/Madam,

We take pleasure in enclosing a revised version of the manuscript entitled: "Peptidyl Nitroalkene Inhibitors of Main Protease (M^{Pro}) rationalized by Computational/Crystallographic Investigations as Antivirals against SARS-CoV-2", to be considered for publication in *Communications Chemistry*.

Please find below the point-by-point responses to the reviewers.

As a general comment, we want to thank the reviewers, not only for their positive comments and global evaluation of our study, but for their suggestions that have been used to improve the quality of the manuscript. In particular, the comments, suggestions and questions raised by the reviewers have been addressed as follows:

Responses to Reviewer 1

Reviewer's 1st comment: "The reversibility of the inhibition by FGA146 has been examined by means of dilution assays (Fig. 11). But the interpretation of these experiments seems unclear because as noticed by the authors "the activity of the Mpro preincubated with FGA146 seems to recover over time, which makes it likely that the compound inhibits its target reversibly (my italics). As a matter of fact, the authors have removed the expression "reversible inhibitors" from the abstract and the final summary. This seems at odds with the following statement from the rebuttal letter "work to confirm the reversibility of the inhibitors has been carried out." Probably, further study would have been required to clarify this important point (why FGA145 and/or FGA147 were not considered for the dilution experiments?), but I believe that there is only weak evidence supporting the reversibility of the proposed inhibitors. In any case, a more clear discussion about this point seems necessary."

Response: Reversibility assays as well as pharmacokinetic studies were carried out with the most active compound, FGA146, it was not considered to carry them out with more compounds of the family due to time limit and resources. We believe that the dilution assays performed over FGA146 clearly demonstrate that the nitroalkene warhead behaves in a reversible fashion against Mpro, as it was the case for other cysteine proteases as we previously have reported (see ACS Med. Chem. Lett. 2016, 1073).

The expression “reversible inhibitors” has been put back in the abstract and in the final summary as suggested by the reviewer.

Response to the reviewer’s comments on computational work: regarding to the computational work, we have reduced substantially the computational work presented in the manuscript, and also new computational work has been performed to address the reviewer concerns (see next). Most of the computational part has been moved to the supplementary information, so any interested reader could follow the subject. We have left in the manuscript the proposed mechanism of nitroalkene inhibition and the optimized structures.

Reviewer’s 2nd comment: “In the revised manuscript, the authors have explored the non-covalent binding step by means of AM1/MM alchemical free energy calculations. In their letter, they claim that “the agreement between the predicted thermodynamics and the measured inhibition equilibrium constants, is dramatically improved”. Thus, the combination of the binding free energy for the E+I->E:I process with the free energy for the chemical E:I->E-I step results now in total ΔG values of -28.9 and -31.7 kcal/mol for FGA146 and FGA147, respectively. It is simply stated in p 13 | 270-271 that “These results are in agreement with the almost equivalent experimentally measured equilibrium KI values” (i.e., around 1-2 μM). However, the calculated energies can be translated into effective KI values of 10-16 and 10-18 μM for FGA146 and FGA147. Apart from the large overestimation of the binding free energy (which raises serious concerns about the adequacy of the QM/MM methods and protocols), it turns out that FGA147 is predicted to be 100-fold more effective than FGA146. Maybe such difference could have been considered to be small, perhaps being comparable with the (undetermined) statistical uncertainty of the free energy calculations and/or in consonance within previous benchmarks of the theoretical methods against validated experimental targets, but no further information or explanation are provided. Therefore, the claim made by the authors about the agreement between experimental and computational data is unsupported.

Response: We agree with the reviewer that further comments should have been added on the energetics results, regarding the associated uncertainty of the employed methods as well as what we wanted to say by “agreement” between experimental and computational results. In this regard, the uncertainty of the exploration of the binding step is not low because we are using a semiempirical method to describe the QM region, while the rest of the process (the chemical steps from E:I to E-I) was explored at DFT/MM level. We agree with the reviewer that a high disagreement is obtained between the predicted KI values and the experimental values reported in Table 1. Consequently, and following the reviewer suggestion, we have removed the study of this step and we have moved to free energy profiles of the chemical steps to the Supporting Information. It is important to point out that uncertainties of ca. 1 kcal/mol in computing activation

or reaction energies in such large and flexible systems with statistical simulations is not unusual, just from the statistics point of view. And we must take into account that this is a two-step mechanism. Keeping this information in mind, the measured KI values for FGA146 and FGA147, which are around 1-2 μM (see Table 1), would correspond to reaction energies of ca. -14 $\text{kcal}\cdot\text{mol}^{-1}$, which is close to the reaction energy of FGA146 (-15.6 $\text{kcal}\cdot\text{mol}^{-1}$) but much larger than the value obtained with FGA147 (-9.8 $\text{kcal}\cdot\text{mol}^{-1}$). We are aware that we still have two problems: 1) there is a disagreement between the almost indistinguishable KI values of FGA146 and FGA-147, as reported in Table 1, and the significant differences of the reaction free energies computationally predicted (from E:I to E-I); and 2) the experimentally determined KI values are measured from E-I to E+I while the binding step (from E+I to E:I) is not take into account in the new version of the manuscript, once we decided to remove this step from the study because of the possible higher uncertainty.

In all, following the reviewer's suggestion, we have removed the study of the binding step and we have moved to free energy profiles of the chemical steps to the Supporting Information. In this revised version of the manuscript, all computational results are based on M06-2X/MM simulations. We have added some comments, giving credit to the obtained mechanism and structures are slowing down the emphasis on the energetics.

Reviewer's 3rd comment: "In their letter (but not in the SI), the authors indicate that they performed 100 ps AM1/MM simulations for calculating the interaction energies (IEs) reported in Figure 6 for the E:I and E-I states. In view of the free energy profiles in Figure 4, the significance of the IEs is very limited. For example, the large difference in the stability of the E:I complexes (8.9 kcal/mol in favor of FGA146), cannot be rationalized in terms of the IEs: FGA147 has larger IEs with P39, D48, H49 N142, M165, H172, Q189, S1 and R4 while only N28, E166 and R186 favor FGA146.

Response: We apologize for not describing into detail the length of the AM1/MM MD simulations carried out to get the pattern of interactions substrate:protein in E:I and E-I states, as shown in old Figure 6. We have improved the text and the SI. However, it is important to note that Figure 6 (now Fig. S14) reports potential energies of interactions (with their errors depicted as error bars) between the substrate and the different residues of the active site (the QM-MM energetic term in the total QM/MM Hamiltonian), and only values higher than 2 kcal/mol are reported (once again, our apologize because we did not mention this point in the caption). They are not total potential energies of the systems (QM + MM + QM-MM) but just the interaction energy term (QM-MM). Thus, a direct comparison with the free energies of E:I and E-I relative to their corresponding solvated separated species E + I (old Figure 4, now Fig. S15) cannot be done, taking into account the size of the systems, their flexibilities, the different computational methods

and the different character of the reported energies. We have added a comment in the text to improve the description of our study and we have moved the figure to the SI, keeping in mind that this is the only results reported in the new version of the manuscript that were not computed at DFT/MM level.

Reviewer's 4th comment: "The same 100-ps AM1/MM MDs were used to obtain the QM/MM snapshots shown in Figures 7 and 8. However, it is well known that 100 ps MD trajectories do not provide significant sampling of the side chain and loop motions so that the comparison with X-ray data is statistically unreliable. Similarly, the relevance of the qualitative flexibility analysis in Figure 8 is very low. Therefore, the structural results from the AM1/MM simulations cannot be accepted for publication. Alternatively, the authors may focus on the comparison of key interatomic distances between the M062X/MM critical structures (E-I in Table S7) and X-ray data. In this respect, I note that TS2 configurations in Figure 5 are missing and they must be included. The relative M062X/MM energies of the optimized structures should be also reported in Figure 5 to assess the weight of thermal effects in comparison with the profiles in Figure 4.

Response: We thank the reviewer for his/her criticisms, which have been used to improve the manuscript. However, we would like to make some comments. First, the structures that appeared in old Figure 7 (now Figure S18) correspond to those derived from the M06-2X/MM simulations. We apologies for the lack of information in the text and in the caption of the figure. Regarding the reviewer's concerns on old Figure 8, we would like to apologize once again for the lack of clarity in the description of the caption. The computational derived structures correspond to those generated from MD simulations with classical force fields, not 100 ps of AM1/MM MD. Anyway, we agree with the reviewer that even 100 ns of MD simulations would not be long enough to explore certain movements of the protein. However, this MD simulations (10 ns of MM MD) were not performed to obtain a full conformational space exploration of the system but to explore, at some extent, the flexibility of the substrate. In fact, a very good overlapping is obtained when superimposing the X-ray structures with the conformational structures of the inhibitor sampled with just these 100 ns MD with classical potentials. As shown in old Figure 7, in old Figure 8 and the geometrical parameters of the tables S7 and S8, the fluctuations of the inhibitors obtained at MM level overlap the single structures optimized at DFT/MM level. Anyway, following the reviewer suggestion, caption of Figure 7 (now Figure S18) has been improved, and Figure 8 has been moved to the Supporting Information (Figure S16). Finally, TS2 was not reported in Figure 5 (now Figure 4) because we considered it was not kinetically relevant. However, following the reviewer suggestion, TS2 is now included in Figure 4. We appreciate the reviewer's suggestion.

Reviewer's 5th comment: "Many more details of the computational protocol have been added to the revised SI (further revision of the English and typo corrections are required; there is even a hyphenated phrase in p. S40). Nevertheless, there are a few points that are still unclear.

Response: We thank the reviewer for his/her comments, which have been used to improve the manuscript, including English and typo corrections. In particular, the answers to his/her specific questions are as follows:

Reviewer's 6th comment: "• Prior to the NEB calculations, it seems that the authors carried out M062X/MM geometry optimizations of the E:I, E-I(-) and E-I configurations and then performed the corresponding NEB interpolations (BTW: this information is missing in the SI). How did the authors select the starting geometries for these calculations? Did they pick up representative MD snapshots according to well-defined criteria? This is indeed a critical point in the QM/MM protocol."

Response: The starting geometry was selected from the MM-MD by visual inspection. Then it was minimized using the semi-empirical xTB Hamiltonian. From this geometry the intermediate and the product were derived by initially applying restraints to the geometries, followed by plain relaxations (using FIRE). Those geometries were then used to set up the initial NEB-xTB/MM by linear interpolation among the three starting points. Once it was converged, the different nodes were used to perform the NEB-M06/MM (thus reducing time by using already optimized structures). As stated in the SI, the maxima were inspected to make sure they were associated with the right transition structures of the chemical process. These M06/MM optimized structures were used to obtain an equally spaced milestones to define the collective variable. We thank the reviewer for pointing it out. The text/supporting information has been improved accordingly.

Reviewer's 7th comment: "• In the response letter, it is noticed (i) "... interaction energies are computed at QM level, despite this decision force to use shorter MD trajectories. Because no restraints were applied to the inhibitor or the protein, the flexibility of the systems are preserved.", (ii) "Regarding the AM1/MM simulations, we used the same conditions as in the DFT/MM, ones but switching to a semiempirical method", and (iii) "Our standard protocol for QM/MM calculations usually keeps frozen the large part of the model lying out of a spherical cutoff around QM atoms". These seem to be rather contradictory statements. It is not clear then whether or not positional restraints were used in the AM1/MM MD simulations. I also note in passing that IEs are AM1/MM values, not just AM1."

Response: AM1/MM MD were used just for evaluating the interactions energies. The free part of the model (20 Å apart of the substrate) was complete flexible, meanwhile the atoms involved in the reactions coordinate (3 in these cases) were constrained using a cartesian tether to avoid losing the reaction coordinate (this is especially important when analyzing transition

geometries). The interaction energy is then evaluated as the difference of the AM1/MM quantum energy and the pure gas phase AM1 energy for a particular geometry, plus the classical Lennard-Jones contribution. Thus, the reported energies include not only the interaction of the polarized wavefunction with the classical charges, but also the polarization energy of the molecular orbital (on going from gas phase to the MM environment). We thank the reviewer for pointing it out. The text/supporting information has been improved accordingly.

Reviewer's 8th comment: “• The authors note that “We have used the same non-bonding conditions (PBC/PME) along all the simulations”. The PME settings for the MM MD calculations are standard, but according to the reference paper of the QMCube driver, the QM atoms are described as zero-charge vdW spheres in the MM engines. How do the authors treat the electrostatic QM···QM interactions across images? “

Response: We are sorry our writing could lead to misinterpretation. For using PME, we should rely on a classical interaction between the MM environment and derived charges for the QM atoms (such as Mulliken charges or fitted ones via ChelpG or Merz-Singh-Kollman). But, we make use of the full interaction between the molecular orbital and the external charges (via monoelectronic integrals), and thus PME cannot be introduced (at least in Gaussian, as far as we know). This is the reason why we introduce an active space of 20 Å around the QM substrate, to maintain a fixed truncation scheme (avoiding this way possible dynamic truncation problems), since those MM atoms are the only ones always allowed to interact with the QM atoms. We thank the reviewer for pointing it out. The text/supporting information has been improved accordingly.

Reviewer's 9th comment: “In their letter, the authors remind me that “additional data can be provided by the authors upon request”. Please note that, as a referee, I already did such request and I do it again here. In my first report, I pointed out that Comm Chem guidelines remark the importance that authors provide enough data or methodological information to help others replicate their work. Of course, this guideline is fully in consonance with the increasing importance of data and method sharing in the field of molecular modelling (<https://doi.org/10.1021/acs.jcim.3c00599> ; <https://doi.org/10.1021/acs.jcim.0c01389>). Thus, I would like to recommend the authors to see how a recent Comm Chem paper makes available datasets that contain scripts, coordinate files, solvent-stripped trajectories, etc. (<https://doi.org/10.1038/s42004-023-01006-0>). Hopefully, they may realize that their statement, “the amount of data that is generated by a computational study such as the one we are presenting here is gigantic, which prevents reporting all them”, is far from being reasonable.

Response: The reviewer is absolutely right that the reproduction of molecular simulations is a requirement in Science. We considered the information we provided (initial PDB structures and

a detailed description of the employed methods with all the required variables), can help others to replicate our calculations. Anyway, following the reviewer's suggestion, we have seen recent papers on CommChem, the CommChem guidelines, and the two JCIM papers commented by the reviewer, to report additional information. In this revised version of our manuscript, supplementary information equivalent to the one reported in the paper of Sangamwar, Wade and coworkers, mentioned by the reviewer (<https://doi.org/10.1038/s42004-023-01006-0>), is provided in Zenodo repository (<https://zenodo.org/uploads/10066288>). In this regard, we noticed that there is a sentence "Materials and all other data are available from the corresponding authors on reasonable request" appears in the "Data availability" section of this recent paper published in CommChem, so we decided to leave it in our manuscript, in case any reader had additional requests. In addition, following the guidelines reported in J. Chem. Inf. Model. 2023, 63, 3227–3229, "If the full trajectories are too large, a selection of clustered structure representing the respective trajectories should be made available in the public repository" (which is our case) we have deposited in the repository, apart from our input files (including topologies, parameters, input scripts, and initial configurations), a representative equilibrated structure extracted from the trajectory.

Responses to Reviewer 3:

Reviewer's 1st comment: "Most of the revisions are fine, however they have kept the statement "All these results combined suggest the viability of employing these compounds as promising drugs against SARS-CoV-2 and new coronavirus that might appear in the future."

Given the poor cellular activity of the compounds (>1 μ M) and the complete lack of measured ADMET data, I can't agree with that statement.

These are interesting ligands, but there is no evidence presented that they are or can be drugs in the future.

The authors make the statement in the rebuttal letter:

"Response: We agree on in vivo data regarding the compounds. Since rodent models need intensive planning and regulations, the data cannot be generated in a short period of time. Pharmacokinetic profiling has been performed as explained above."

In vivo pharmacokinetic data has not been obtained, they haven't done the study needed to demonstrated free exposure over the cellular EC90. The statement made "Since rodent models need intensive planning and regulations the data cannot be generated in a short period of time." Is factually untrue. There are numerous CROs who will conduct these studies on supply of material, it is a funding and contracting issue. I appreciate they may need to fund the study and

supply appropriate material - but it is simply inadequate to make this excuse if they want to make that assertion.

The literature is plagued with ligands claimed to be drugs with no evidence of exposure, this creates a false hope and vastly underestimates the scientific challenges of actually generating new therapeutic agents.

Either they need to generate the data to support the assertion "All these results combined suggest the viability of employing these compounds as promising drugs against SARS-CoV-2 and new coronavirus that might appear in the future." or withdraw it.

Response: We agree with the reviewer that we cannot claim they are drugs. All the sentences claiming they are drugs have been either removed or changed:

In the abstract, the sentence "(...) and that inhibitors **FGA145**, **FGA146** and **FGA147** prevent infection becoming promising drugs against SARS-CoV-2." has been changed to "(...) and that inhibitors **FGA145**, **FGA146** and **FGA147** prevent infection against SARS-CoV-2."

In the conclusions, the sentence "All these results combined suggest the viability of employing these compounds as promising drugs against SARS-CoV-2 and new coronavirus that might appear in the future." has been removed.

We look forward to hearing from you, hoping this revised version of the manuscript will be suitable for the readers of your journal.

Yours sincerely,

Florenci V. González

REVIEWERS' COMMENTS:

Reviewer #1 (Remarks to the Author):

After the second revision process, I consider that my previous concerns have been clearly addressed. The final version of the manuscript has been substantially shortened and improved. Therefore, I recommend this contribution for publication. Let me also thank the authors for their work.

Castelló, December 11th, 2023

Dear Sir/Madam,

We take pleasure in enclosing a revised version of the manuscript entitled: “Peptidyl Nitroalkene Inhibitors of Main Protease rationalized by Computational and Crystallographic Investigations as Antivirals against SARS-CoV-2” , to be considered for publication in *Communications Chemistry*.

Please find below the point-by-point responses to the reviewers.

As a general comment, we want to thank the reviewers, not only for their positive comments and global evaluation of our study, but for their suggestions that have been used to improve the quality of the manuscript. In particular, the last comment made by the Reviewer 1:

Responses to Reviewer 1

Reviewer #1 (Remarks to the Author):

Reviewer’s comment: After the second revision process, I consider that my previous concerns have been clearly addressed. The final version of the manuscript has been substantially shortened and improved. Therefore, I recommend this contribution for publication. Let me also thank the authors for their work.

Response: We appreciate the reviewer’s comment and we are thankful to her/him for the time spent to revise the work.

We look forward to hearing from you, hoping this revised version of the manuscript will be suitable for the readers of your journal.

Yours sincerely,

Florenci V. González